

# Comparison of variables from ocean, sea ice and atmosphere models as forcing data for iceberg drift and deterioration models in the Barents Sea in 2010-2014 and 2020-2021 (Part I).

Lia Herrmannsdörfer[1], Raed Khalil Lubbad[1], and Knut Vilhelm Høyland[1]

[1]Norwegian University of Science and Technology, Trondheim, Norway

**Correspondence:** Lia Herrmannsdörfer (lia.f.herrmannsdorfer@ntnu.no)

**Abstract.** Numerical models of ocean, sea ice and atmosphere supply a wide range of information in the Arctic that are difficult to observe otherwise. Model disagreements emphasise the need to evaluate the suitability of the models for individual applications. This study compares selected ocean, sea ice and atmosphere variables from the models Topaz4b, Barents-2.5, ERA5 and CARRA in the Barents Sea during the years 2010-2014 and 2020-2021. The same data is used in the sequel paper

(Herrmannsdörfer et al., 2024) to force simulations of iceberg drift and deterioration and to examine the impact of varied forcing on the iceberg simulations. Comparing Topaz4b and Barents-2.5, it is evident that sea ice is more extensive (larger sea ice concentration, thickness and southward extent) and sea surface temperatures are lower in Barents-2.5 with clear differences in the seasonal and spatial characteristics. Further, sea surface and sea ice drift speeds are larger in Barents-2.5, especially in shallow waters and the sea ice edge. On the side of atmospheric models, CARRA exhibits slightly larger 10m wind speeds

over open water while ERA5 show larger wind speeds over icy water. Those similarities and differences could partly be traced back to similarities and differences in spatial and temporal resolution, model setup, assimilated data and relations between the models. Despite fundamental difference in data assimilation, Barents-2.5 hindcast and forecast showed high similarity for some variables. The large occurrence of sea ice and its deviating representation in the models indicate large relevance for the iceberg pathways in the Barents Sea.

## 1   Introduction

Numerical models are a powerful tool to describe regions of the Earth (e.g. the Arctic) and time periods (e.g. early 20th century) with little observations. They provide spatially and temporally consistent information, that are constraint to the physics and observations.

Various ocean, sea ice and atmosphere models describe the Arctic, however the disagreement between the models can be large for some variables. For example, in connection to the Ocean ReAnalyses Intercomparison Project (ORA-IP), Chevallier et al. (2017) studied 15 ocean and sea ice reanalyses and found similarities and differences in the model physics, data assimilation methods, forcing and resulting representation of Arctic sea ice. Notz and Community (2020) analysed Arctic sea ice in the more recent Coupled Model Intercomparison Project (CMIP6) and found significant disagreement in the representation of





sea ice area in different climate models. Jakobson et al. (2012) compared 5 atmospheric reanalyses, revealing differences in e.g. air temperature, water content and wind, and described the strengths and weaknesses of the models for different variables and model layers. Lindsay et al. (2014) compared seven atmospheric reanalyses in the Arctic for a wide range of variables (e.g. radiative fluxes, precipitation and temperature) and analysed the impact of their varied output as forcing to ocean and sea ice models. In presence of large model differences, some models may be more suitable for some variables and settings.

The suitability of the model depends on the application, the study region and time. Nevertheless, other factors such as ease of access, ownership, convenience and previous use often govern the choices in practice. Often, little effort is spent on evaluating the suitability of the different available models and data products. This is inexcusably justified by the scarcity of information which can obstruct a meaningful comparison. While some ocean, sea ice and atmosphere models and variables are documented

well (e.g 10m wind in ERA5 (ECMWF, 2016)), information on a wide range of details (e.g. model equations, data assimilation approach) is missing (e.g. sea water surface velocity in Topaz4b (MDS, 2023)).

This paper compares atmospheric, ocean and sea ice variables from different numerical models in the domain of the Barents Sea for the years of 2010-2014 and 2020-2021. Thus, the spatial and temporal differences of the variables in the different

models are examined and compared with knowledge from literature. The results of this study will be used in the sequel paper (Herrmannsdörfer et al., 2024) to estimate the impact of varied environmental forcing on simulations of iceberg drift and deterioration in the Barents Sea. The choice in study time and models is based on the availability of model variables for the Barents Sea for two sets of differently resolved atmosphere-, ocean- and sea ice-models. The comparison is focused on the variables of interest and aspects relevant to the simulations of iceberg drift and deterioration. However, the comparison may also be used in

other geophysical studies that enhance the understanding of the domain of the Barents Sea, or to evaluate the suitability of the atmosphere, ocean and sea ice models for other applications (e.g. statistics of sea ice conditions for shipping operations).

The analysis herein includes the variables of 10 m wind velocity ($v_\mathrm{a}$), sea water surface velocity ($v_\mathrm{w}$), sea surface temperature ($SST$), sea ice fraction ($CI$), sea ice thickness ($h_{si}$) and sea ice drift velocity ($v_\mathrm{si}$).


The analysis is performed in 3 steps. First, an overview of the selected atmosphere, ocean and sea ice models, their variable quality and model relations is given (Sect.2). A more extended comparative study of those models is given in the Appendix (Sect.A). It compares the model setup, representation and quality of the variables of interest based on available information in the open literature. Second, the output of the atmosphere, ocean and sea ice models is compared statistically for the variables

of interest in the analysis period and domain (Sect.3.1). Last, the same data is compared statistically, but this time along the main iceberg pathways in the Barents Sea (Sect.3.2). The results are discussed in the light of previous findings and the iceberg simulations in Herrmannsdörfer et al. (2024) (Sect.4) before the main conclusion are drawn.





## 2 Data

In this section, a short overview on the used atmosphere, ocean and sea ice models, their variable quality and relations is given

(Table 1, Fig. 1). A detailed comparison of the models is given in the Appendix (Sect.A). The described atmosphere-, ocean-
and sea ice-models treat the relevant variables of this study as prognostic variables (Duarte et al., 2022; Hersbach et al., 2020;
Schyberg et al., 2023). The models have different temporal and horizontal resolution, different bathymetry (topography) and
representation of the coastline.

### 2.1 Ocean and sea ice models

#### 2.1.1 Arctic Ocean Physics Reanalysis (Topaz)

The Arctic Ocean Physics Reanalysis (ARCTIC_MULTIYEAR_PHY_002_003) is an ocean and sea ice reanalysis for the
Arctic based on the fully coupled ocean and sea ice model Topaz4b, from 1991 to 2021 (Sakov et al., 2012; MDS, 2023) (Table
1). It is run by the Nansen Environmental and Remote Sensing Center (NERSC) and Met-Norway (Norwegian Meteorological

Institute), and is provided by Copernicus Marine. It is referred to as *Topaz*, in the following.

#### 2.1.2 Barents-2.5

Barents-2.5 forecast is a short-time, high resolution regional forecasting system for ocean and sea ice in the Barents Sea
from 2020 to present (MET-Norway, a; Röhrs et al., 2023) (Table 1). It is produced and provided by Met-Norway. Due to
its comparably high resolution, Barents-2.5 is used for computing trajectories of pollutants and drifting objects, that shall be

retrieved in emergency situations and for planning, for example in the framework OpenDrift (Röhrs et al., 2023; Idžanović
et al., 2023). This study uses the forecast version v.1 from in 2020 and 2021, that was updated to a full ensemble prediction
system (EPS, v.2) in 2022 (Röhrs et al., 2023). In addition, a preliminary version of the Barents-2.5 hindcast (Idžanović et al.,
2024) is used from 2010 to 2014.

#### 2.1.3 Similarities, differences and variable quality in Topaz and Barents-2.5

Topaz and Barents-2.5 have a similar model setup, in which the ocean and sea ice component are fully coupled (MDS, 2023;
Duarte et al., 2022). Topaz and the Barents-2.5 forecast assimilate satellite-based products of $SST$ and $CI$, however at differ-
ent resolutions and from different sources. Topaz also assimilates $h_{si}$ from October-April (MDS, 2023; Hackett et al., 2022;
Röhrs et al., 2023). Due to the general lack of observations of sea water velocity, neither model assimilates such information
(MDS, 2023; Röhrs et al., 2023). The Barents-2.5 hindcast does not assimilate observations in the domain (Idžanović et al.,

85  2024).



**Table 1.** Overview on ocean, sea ice and atmosphere models used in this study.

| Model | Arctic Ocean Physics Reanalysis (Topaz) | Barents-2.5 Hindcast & Forecast | Global atmospheric re-analysis (ERA5) | Copernicus Arctic Regional ReAnalysis (CARRA) |
|---|---|---|---|---|
| What | Ocean and sea ice | Ocean and sea ice | Atmosphere | Atmosphere |
| Type | Reanalysis | Hindcast, Forecast (EPS) | Reanalysis | Reanalysis |
| Horizontal resolution | 12.3 km | 2.5 km | 31 km | 2.5 km |
| Temporal resolution | daily to monthly | hourly | hourly | 3 − hourly |
| Domain | Arctic Ocean north of $50°N$ | Barents Sea | Global | Barents Sea, Greenland |
| Time | 1991-2022 | 2010-2022, 2020-2021 (non-EPS), 2022-present (EPS) | 1950-present | 1990-present |
| Producer | NERSC, MET Norway | MET Norway | ECMWF | MET-Norway |
| Supplier | Copernicus Marine | MET Norway | C3S | C3S |
| Literature Reference | Sakov et al. (2012) MDS (2023) | Röhrs et al. (2023) MET-Norway (b), MET-Norway (a) | Hersbach et al. (2020) Hersbach et al. (2023) | e.g. Køltzow et al. (2019) Schyberg et al. (2023) |
| Link to product | https://doi.org/10.48670/moi-00007 | https://thredds.met.no/thredds/catalog/romshindcast/barents2500_2010/catalog.html, https://thredds.met.no/thredds/catalog.html | https://doi.org/10.24381/cds.adbb2d47 | https://doi.org/10.24381/cds.713858f6 |
| Product | Subset of daily surface product | Subset of shortest lead time 2010-2014 hindcast and 2020-2021 (non-EPS) forecast | Subset of product on single levels | Subset of eastern domain product on single levels |
| Variables[a] | $SST$, $\boldsymbol{v}_\mathrm{w}$, $h_{si}$, $CI$, $\boldsymbol{v}_\mathrm{si}$ | $SST$, $\boldsymbol{v}_\mathrm{w}$, $h_{si}$, $ci$, $\boldsymbol{v}_\mathrm{si}$ | $\boldsymbol{v}_\mathrm{a}$ | $\boldsymbol{v}_\mathrm{a}$ |
| DA[b] | $SST$, $CI$, $h_{si}$, $\boldsymbol{v}_\mathrm{si}$ | $SST$, $CI$ | $\boldsymbol{v}_\mathrm{a}$ | no |
| Forcing by other used models | Forced at water surface by ERA5 atmospheric fields | Hindcast: initial conditions & lateral boundaries from Topaz reanalysis, surface by ECMWF ENS (same model origin as ERA5). Forecast: lateral boundaries from Topaz forecast, surface by AROME Arctic (same model origin as CARRA) | No | Forced by ERA5 at lateral boundaries |

[a] Variables (used in this study): sea surface temperature ($SST$), sea surface water velocity ($\boldsymbol{v}_\mathrm{w}$), sea ice thickness ($h_{si}$), sea ice concentration ($ci$), sea ice velocity ($\boldsymbol{v}_\mathrm{si}$), 10m wind velocity ($\boldsymbol{v}_\mathrm{a}$)

[b] DA: Data assimilation. In this case, assimilation of observations of the used prognostic variables



In validation of the variables of interests from Topaz, Xie et al. (2017) and Xie and Bertino (2022) compared to a range of (mostly satellite-based) products in the Arctic between 1990 and 2020. The assessment showed a general underestimation of $SST$, $CI$, the sea ice area with $CI > 15\%$ and $h_{si}$, while on the other hand it showed an overestimation of sea ice drift. Topaz also shows a seasonal cycle with too small $SST$ and $CI$ but too fast drift in winter and too warm $SST$ but underestimated drift in summer. $CI$ is represented well in summer. $h_{si}$ is underestimated in autumn. In addition, the sea ice extends and declines too fast in spring and autumn. Large errors occur in the northern part of the domain for the sea ice drift speed and in representing the sea ice edge. $SST$ showed a large negative bias in the Barents Sea and the Eurasian Basin, and large positive bias along the shelf-edge of southern Svalbard, Spitsbergen Bank, Storfjorden- and Bjørnøya Trough. Some of these biases are related to a seasonally varying number of assimilated observations, a spatially varying observational error, the simulated circulation of Atlantic water inflow and the topographic steering. For the sea water surface velocity (uppermost sea water layer from 0 to 1m, (MDS, 2023)), no quality control is at hand.

Validation is available for the Barents-2.5 hindcast from 2010 to 2018 (Idžanović et al., 2024) and the an EPS version of the Barents-2.5 forecast in 2022 (Röhrs et al., 2023) and from 2022 to present (MET-Norway, 2024). These information are used, as no direct validation has been performed on the exact forecast product used in this study. The validation of Barents-2.5 hindcast and EPS forecast compared $SST$ and $CI$ with satellite-based observations in the Barents Sea in the years of 2010 to 2018 and during short time periods in 2021-2022 and showed general overestimation of sea ice concentrations and underestimation of $SST$. Those errors are larger in the Barents-2.5 hindcast than in Topaz, especially for $CI$ in the first half of the year. The validation described the Barents-2.5 forecast of $SST$, $CI$ and sea ice drift as skilful (in periods with enough data assimilation), however also revealed large mismatches of $SST$ in the marginal ice zone. The forecast model state drifts off and looses its skill within few days without assimilation of sea ice information. The sea water surface velocity in Barents-2.5 is responsive to air pressure, tides and small scale processes (Röhrs et al., 2023), and is therefore used in the OpenDrift framework (Dagestad et al., 2018). However, in the comparison to land-based radar data during some month of 2021, speed and direction show low predictive skill (Idžanović et al., 2023). Water speed and direction are also varying on temporal- and spatial scales, due to the lack of observations, information about observation and model error and the chaotic nature of the system (Röhrs et al., 2023). However, Röhrs et al. (2023) describes how low skill water surface velocity may provide value due to the response on topography, tides and wind. In conditions when the water motion is mainly wind-driven (where the initial conditions are less relevant), some skill is associated with the Barents-2.5 output, forced by the high resolution atmospheric data (AROME-Arctic) (Idžanović et al., 2023).





### 2.2 Atmospheric models

#### 2.2.1 Global atmospheric reanalysis (ERA5)

ERA5 is a global atmospheric reanalysis from 1940 to present (Hersbach et al., 2020, 2023) (Table 1). It is provided by the European Centre for Medium-Range Weather Forecasts (ECMWF) on the Copernicus Climate Data Store (CDS).


#### 2.2.2 Copernicus Arctic Regional ReAnalysis (CARRA)

Copernicus Arctic Regional ReAnalysis (CARRA) is a local high resolution atmospheric reanalysis for the Barents Sea and Greenland with particular focus on the interaction with the cryosphere, from September 1990 to present (Schyberg et al., 2023; Copernicus Climate Change Service (C3S), 2023) (Table 1). It is produced by Met-Norway and provided on CDS.


#### 2.2.3 Similarities, differences and variable quality in ERA5 and CARRA

ERA5 assimilates a range of wind observations (Hersbach et al., 2020). CARRA assimilates information about the surface and upper atmosphere from ERA5, but does not assimilate the 10m wind (Yang et al., 2020b). ERA5 and CARRA are similar over open ocean, where their model errors are small (Hersbach et al., 2020; Køltzow et al., 2022; Giusti, 2024). The validation
against selected SYNOP observations in the North-Atlantic and European Arctic during winter and summer months of 2010-2019 in Giusti (2024) showed under- (ERA5) and over-estimation (CARRA) of the 10m wind speed. Thereby, ERA5 has larger model, observational- and representativity- and random error. ERA5 also underestimates extreme wind speeds around Svalbard (Køltzow et al., 2022). The model errors vary seasonally and spatially, and are largest (smallest) in winter (summer) (Giusti, 2024). The spatial (sub-grid) variability is large for both models (Køltzow et al., 2022). Both models assimilate and prescribe
satellite-based information of $SST$ and $CI$ (Hersbach et al., 2020; Køltzow et al., 2022), and assume level sea ice. As a result, both models overestimate near surface winds over multi-year sea ice (Personal communication at CARRA workshop, 21 Sep 2023). However, CARRA has added value over sea ice, due to its improved physical parametrisation and assimilation of higher resolution satellite observations of sea ice (Giusti, 2024). Thereby, the 10m wind speed in CARRA has improved temporal and spatial variability, is more accurate over complex topography and has added value over ERA5, in general (Giusti, 2024;
Køltzow et al., 2019, 2022).

### 2.3 Model relations

The model's data assimilation (DA) comprises observational data and output from other models. The assimilated data is partly mixed into the model state, used as forcing at the interface of atmosphere with ocean and sea ice (surface), or as forcing at the lateral boundaries. Thus, the atmosphere, ocean and sea ice models are not independent of each other. The model relations
are visualised in Fig. 1. In summary, ERA5 is independent of the other environmental models, but supplies information of the





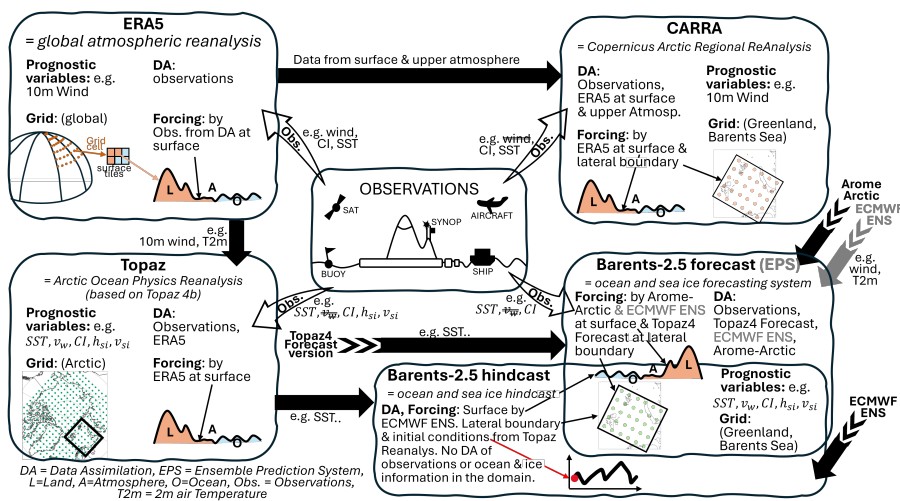

**Figure 1.** The atmosphere, ocean and sea ice models Topaz, Barents-2.5, ERA5 and CARRA, their relations, grid and data assimilation.

surface to Topaz (Hackett et al., 2022) and information of the surface and upper atmosphere to CARRA (Yang et al., 2020b). The Barents-2.5 forecast is forced at the surface by AROME Arctic (Duarte et al., 2022; Röhrs et al., 2023), which has a similar model ancestor to CARRA. The Barents-2.5 forecast is also forced at the lateral boundaries by a forecast version of Topaz. The Barents-2.5 hindcast adapts its initial conditions and forcing at the boundaries from the Topaz version used in this study (Idžanović et al., 2024). The Barents-2.5 hindcast is also forced by ECMWF ENS, which has a similar model ancestor than ERA5.

## 3 Analysis

### 3.1 Statistical analysis of selected model variables in the Barents Sea

The data description (Sect.2) and the detailed comparative study of the models in the Appendix (Sect.A) shed light on the general similarities and differences between the selected ocean, sea ice and atmosphere models. However, validation or direct comparison is not readily available for all used products. Furthermore, regional phenomena and multi-year variability may cause large deviation to the described cases. Following, variables of ocean, sea ice and atmosphere from Topaz, Barents-2.5, ERA5 and CARRA are compared directly for the periods 2010-2014, 2020-2021 over the region of the Barents Sea.

### 3.1.1 Data pre-processing and methods of comparison

Table 1 shows the variables, availability and used product of the Topaz, Barents-2.5, ERA5 and CARRA data. Some variables require pre-processing. As such, CARRA components of wind velocity are calculated from the supplied CARRA wind speed and direction. A local grid rotation is performed for the velocity variables of Barents-2.5 given in model grid direction.



As a consequence, all used velocity components are oriented in the longitudinal (u) and latitudinal (v) direction. Further-more, the 10m wind is masked for grid cells with at least 50/75% water surface according to the model's land-sea-mask in

ERA5/CARRA, as this analysis is limited to ocean and sea ice surfaces. The preliminary version of the Barents-2.5 hindcast used in this study was further processed by projecting the staggered production grid onto a common grid, matching the new supplied product, MET-Norway (b).

To enable element-wise comparison between the different environmental datasets (in this case grid cell and time step-wise),

the temporal and spatial resolution is subsetted and projected onto a common frequency and grid. The choices in temporal and spatial projection are made to match the assimilation of environmental data into the model for iceberg drift and deterioration in Herrmannsdörfer et al. (2024).

A temporal subset of the datasets is taken for the years of 2010-2014 and 2020-2021, as these were the only years available

in all four atmosphere, ocean and sea ice models upon the time of data pre-processing. The temporal frequency of the datasets is resampled, as shown in Fig. 2. The variables from ERA5 and Barents-2.5 are downsampled to $2-\mathrm{hourly}$ data by discarding odd hours. Topaz is upsampled to $2-\mathrm{hourly}$ data by repeating the $\mathrm{daily}$ value at $2-\mathrm{hourly}$ frequency until the next 0UTC time step. CARRA is sampled $2-\mathrm{hourly}$ by repeating the last available $3-\mathrm{hourly}$ value.

A spatial subset of the datasets is selected, that covers the Barents Sea and is as similar as possible in all datasets (Fig. 3a). Note that the models have different grids types and grid rotation (Fig. 3b), thus the subsets cover slightly different areas. Note also difference in the models land-sea masks in Fig. 3a).

Further, the datasets with low horizontal resolution are projected onto the grid of the higher resolution model by nearest

neighbour interpolation, as shown in Fig. 4a and b. This means that for every grid cell of the Barents-2.5 and CARRA grid, the information of wind, sea ice and sea water are adapted from the nearest grid cell of Topaz and ERA5. The distance between the grid cells of different model grids is determined by the distance of the grid cell centres. As a result, we receive the Topaz and ERA5 variables on the same grid as Barents-2.5 and CARRA, allowing for an element-wise comparison.

The differences between the ocean, sea ice and atmosphere models are presented, as map plots and timeseries. The spatial differences are presented as map plots of $2.5\,\mathrm{km}$ resolution, averaged over the time periods of 2010-2014 and 2020-2021. The temporal differences are presented in a time series of $2-\mathrm{hourly}$ resolution, showing the spatial average over the spatial domain of the Barents Sea. They also show the variable range from 5th to 95th percentile in the domain, that expresses the variability in the domain, excluding extreme values. In addition, the average model difference over time and space and relevant

absolute values are given, to estimate the relative importance of absolute values and model deviations.





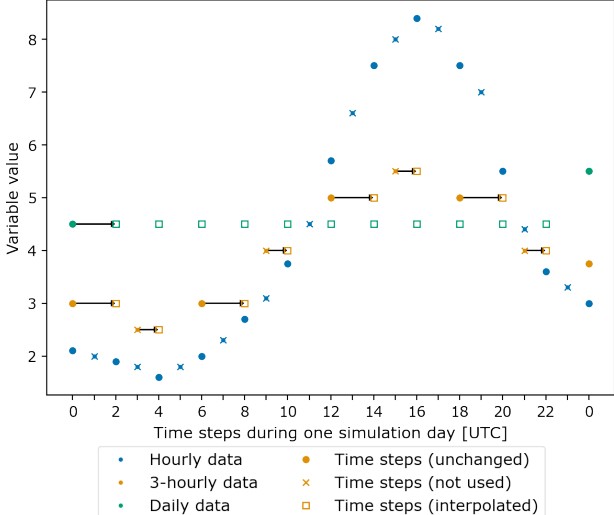

**Figure 2.** $2-$hourly resampling of the original temporal resolution of Topaz (daily, green), Barents-2.5 (hourly, blue), ERA5 (hourly, blue) and CARRA ($3-$hourly, orange). Time steps are used unchanged (dots), discarded (crosses) and are sampled from a previous time step (arrows, square). In detail, hourly data from ERA5 and Barents-2.5 (blue) is downsampled $2-$hourly (dots), discarding odd hours (crosses). $3-$hourly data from CARRA (orange) is used unchanged at the time steps 0,6,12 and 18 (dots), and sampled from the previous available time step at the remaining hours (arrows, squares). The daily data from Topaz (green) is repeated $2-$hourly (squares) until the next day.

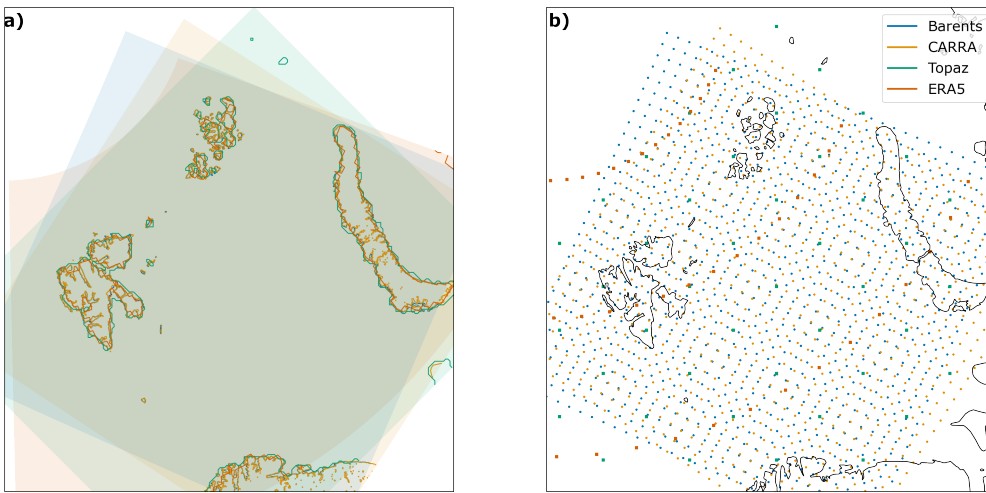

**Figure 3.** a) Spatial subset, representation of the coastlines and b) model grids of Topaz (green), Barents-2.5 (blue), ERA5 (red) and CARRA (orange). Note that b) only shows every 20th grid point in x and y direction to increase visibility.





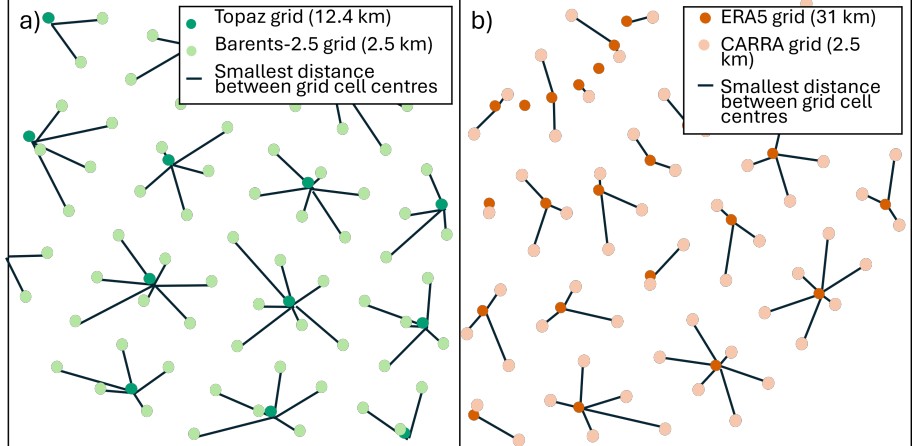

**Figure 4.** Spatial resampling methods for the projection between grids of different horizontal resolution, rotation and grid type In detail, a) Topaz (12.3 km km, dark green) is projected on the Barents-2.5 grid (2.5 km, light green) and b) ERA5 (31 km, red) is projected onto the CARRA grid (2.5 km, orange).

### 3.1.2 Ocean and sea ice variables

*Sea surface temperature*

Figures 5 and 6 show the results of the comparison between the sea surface temperature (SST) variable from Topaz and Barents-2.5. The Figures expose spatial and temporal differences between the two models. Largest $SST$ differences in the study area, are visible along the west coast of Svalbard, Storfjorden Trough, Bjørnøya Trough and Hopen Trench, with larger $SST$ in Topaz (Fig.5a). In contrast. Topaz has slightly lower $SST$ in the northern part of the Barents Sea. Figure 5 (b,c) and 6 show that the difference of $SST$ in Topaz and Barents-2.5 has a seasonal cycle. In the summer and autumn, Topaz provides higher $SST$ than Barents-2.5, and the $SST$ difference has slightly higher variability over most of the study area. In the winter and spring, Topaz provides lower $SST$ than Barents-2.5 in the northern part of the study and higher $SST$ (compared to Barents-2.5) in the southern part of the study area, leading to a small difference between the ocean models, averaged over the study area and the season. Both models reveal a similar multi-year variability. The average $SST$ difference is $0.55°\mathrm{C}$ (larger in Topaz) in the Barents Sea in the study periods 2010-2014 and 2020-2021 (Table 2). Thereby, the mean model difference varies in the two periods, with $0.48°\mathrm{C}$ (2020-2021) and $0.57°\mathrm{C}$ (2010-2014, Table 2). Further analysis (not shown) yielded that the spatial patterns are similar in those time periods.

*Sea surface velocity*

The Fig.7 and 8 show the results of the comparison between the sea water surface velocity ($v_w$) variable from Topaz and Barents-2.5. The water speed in Topaz and Barents-2.5 shows spatial and temporal differences (Fig.7) and 8). Spatial differences are present around the coastlines of the mainland Europe, archipelagos and islands in the Barents Sea (Fig.7a). We



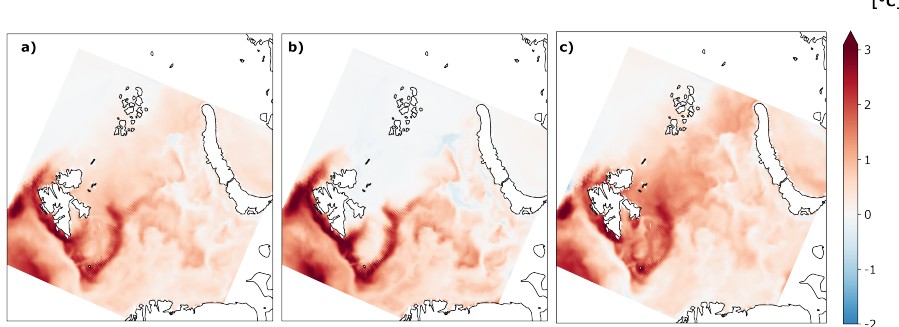

**Figure 5.** Map of sea surface temperature difference (Topaz-Barents2.5, °C) in the Barents Sea, averaged for the a) years 2010-2014 and 2020-2022, b) winter and spring month (Dec-May) and c) summer and autumn (Jun-Nov) of these years. The comparison is performed on a horizontal resolution of 2.5 km, with Topaz projected onto the Barents-2.5 grid.

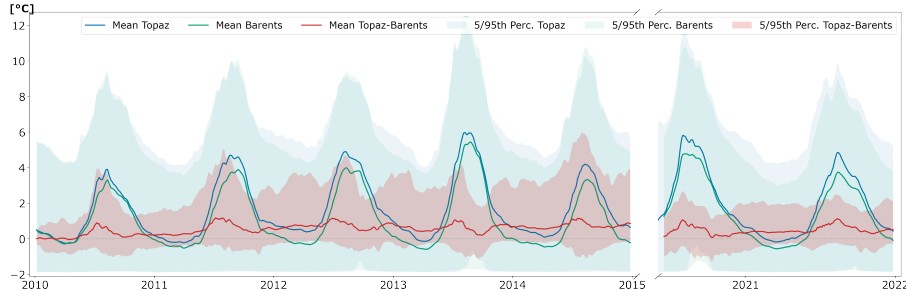

**Figure 6.** Time series of spatial average sea surface temperature [°C] for Topaz (blue line), Barents-2.5 (green line) and their difference (Topaz-Barents2.5, red line) for the analysis time period 2010-2014 and 2020-2022. The shaded areas show the variability of the temperatures (blue/green) and its deviation between Topaz and Barents-2.5 (red) in the domain, excluding the extreme values (5th and 95th Percentile). The comparison is performed with a temporal resolution of 2 hours on the temporally resampled datasets (Sect.3.1.1). A 10 days rolling average is applied.

highlight the large differences around Svalbard, the islands between Svalbard and Franz-Josef-Land, around Spitsbergen Bank and Central Bank. In those regions, Barents-2.5 has larger average water speeds than Topaz. Topaz has larger water speeds, compared to Barents-2.5, towards the Eurasian Basin. Following, the spatial average, water speed is larger in Barents-2.5, compared to Topaz (Fig.8). Figure 8 also exhibits a seasonal cycle in the water speed difference of Topaz and Barents-2.5 and its variability in the domain. In summer and autumn, water speeds, their difference in Topaz and Barents-2.5 and the variability

in the domain are largest. In spring, the speeds, model differences and spatial variability are in smallest. The seasonal cycle can also be seen in the seasonal spatial ocean model difference (Fig.7b,c), however spatial patterns are conserved throughout the year. Further analysis revealed a lower sea water surface speed in Topaz ($-0.047 \, \mathrm{m \, s^{-1}}$), compared to Barents-2.5, averaged over the Barents Sea and the years of 2010-2014 and 2020-2021 (Table 2). The model differences are slightly larger in the



**Table 2.** Mean differences of ocean, sea ice and atmosphere variables in Topaz, Barents-2.5, ERA5 and CARRA. The variables are sea surface temperature (SST), sea surface speed ($v_w$), sea ice concentration ($CI$), sea ice thickness ($h_{si}$), percentage of grid cells containing $CI > 15\%$, light sea ice ($CI > 15\%$, excluding heavy sea ice) and heavy sea ice ($CI > \leq 90\%$, $h_{si} > h_{min}$), sea ice drift speed ($v_{si}$) and 10m wind speed ($v_a$). The variables are averaged over the study area of the Barents Sea and the years 2010-2014 and 2020-2021. The comparison is performed on the temporally and spatially resampled datasets (see Sect.3.1.1), at a resolution of 2 hours.

| Variable | $\Delta(Topaz - Barents2.5)$ | 2010-2014 | 2020-2021 |
|---|---|---|---|
| $\varnothing SST$ [°C] | +0.55 | +0.57 | +0.48 |
| $\varnothing v_w$ [m s$^{-1}$] | -0.047 | -0.045 | -0.052 |
| $\varnothing CI$ [%] | -10 | -13 | 0 |
| $\varnothing h_{si}$ [m] | -0.23 | -0.30 | 0 |
| % $CI > 15\%$ | -6.54 | -12.79 | +9.06 |
| % light sea ice | -0.26 | -2.34 | +4.93 |
| % heavy sea ice | -6.27 | -10.44 | +4.14 |
| $\varnothing v_{si}$ [m s$^{-1}$] | -0.068 | -0.069 | -0.067 |
| Variable | $\Delta(ERA5 - CARRA)$ | 2010-2014 | 2020-2021 |
| $\varnothing v_a$ [m s$^{-1}$] | -0.0007 | +0.02 | -0.05 |

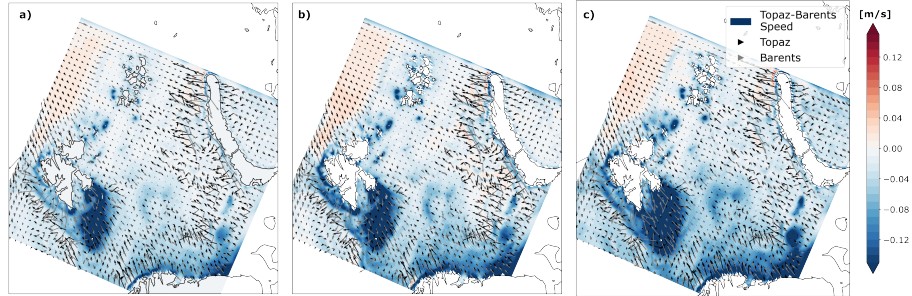

**Figure 7.** Map of sea surface speed difference (Topaz-Barents2.5, colours, m s$^{-1}$) and sea surface current direction (arrows) in the Barents Sea, averaged over a) the analysis time period 2010-2014 and 2020-2022, b) winter and spring month (Dec-May) and c) summer and autumn (Jun-Nov) of these years. The comparison is performed on a horizontal resolution of 2.5 km, with Topaz projected onto the Barents-2.5 grid.

period of 2020-2021 ($-0.052$ m s$^{-1}$), compared to the period of 2010-2014 ($-0.045$ m s$^{-1}$).


### *Sea ice concentration and thickness*

The differences of sea ice concentration ($CI$) and thickness ($h_{si}$) in Topaz and Barents-2.5 are shown in the Fig.9 and 10. The variables show considerable spatial and temporal differences. Throughout most of the Barents Sea, $CI$ and $h_{si}$ are larger in



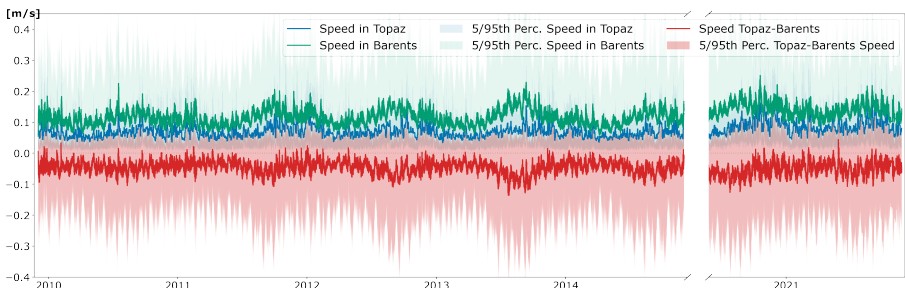

**Figure 8.** Time series of spatial average sea surface speed $[\mathrm{m\,s^{-1}}]$ in Topaz and Barents-2.5 (blue, green) and their difference (Topaz-Barents2.5, red) in the Barents Sea for the analysis years 2010-2014 and 2020-2022. The shaded areas show the variability of speeds and the difference in Topaz and Barents-2.5 in the domain, excluding the extreme values (5th and 95th Percentile). The comparison is performed with a temporal resolution of $2\,\mathrm{hours}$ on the temporally resampled datasets (Sect.3.1.1).

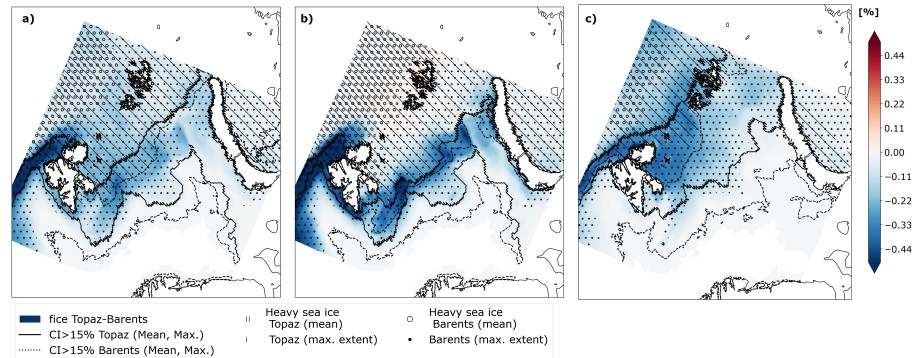

**Figure 9.** Map of sea ice concentration difference (Topaz-Barents2.5, colours), average and maximum extent of regions with sea ice concentration $CI > 15\%$ in Topaz and Barents-2.5 (solid and dotted line), and average and maximum extent of regions with heavy sea ice $CI \geq 90\%$, $h_{si} > h_{min}$ in Topaz and Barents-2.5 (line and point hatches). The variables are temporal averages over the a) the years 2010-2014 and 2020-2022, b) winter and spring month (Dec-May) and c) summer and autumn (Jun-Nov) of these years. The comparison is performed on a horizontal resolution of $2.5\,\mathrm{km}$.

Barents-2.5, compared to Topaz (Fig.9 and 10). Largest differences between the sea ice models are present around the Fram Strait, northern part of Svalbard and Franz-Josef-Land. Large $h_{si}$ difference are also visible between Svalbard and Novaya Zemlya. Further analysis indicates a $10\%$ larger $CI$ and $0.23\,\mathrm{m}$ larger $h_{si}$ in Barents-2.5, compared to Topaz, averaged over the Barents Sea and the years 2010-2014 and 2020-2021 (Table 2). The model differences are large for the years of 2010-2014 ($-13\%\,CI$, $-0.3\,\mathrm{m}\,h_{si}$), but no systematic difference is found in 2020-2021.



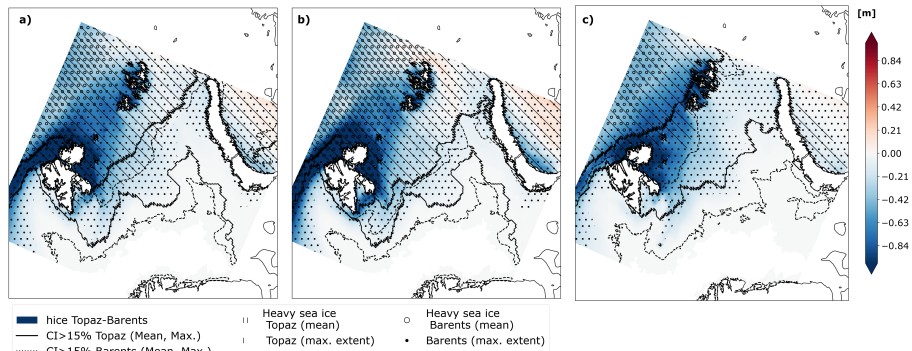

**Figure 10.** Map of sea ice thickness difference (Topaz-Barents2.5, colours), average and maximum extent of regions with sea ice concentration $CI > 15\%$ in Topaz and Barents-2.5 (solid and dotted line), and average and maximum extent of regions with heavy sea ice $CI >= 90\%$, $h_{si} > h_{min}$ in Topaz and Barents-2.5 (line and point hatches). The variables are temporal averages over the a) the years 2010-2014 and 2020-2022, b) winter and spring month (Dec-May) and c) summer and autumn (Jun-Nov) of these years. The comparison is performed on a horizontal resolution of $2.5\,\text{km}$.

Further, we define categories of sea ice, namely light sea ice with $CI > 15\%$ (excluding heavy sea ice) and heavy sea ice with $CI \geq 90\%$ and the strength criterion $h_{si} > h_{min}$ with

$$h_{min} = \frac{13000}{20000 * exp(-20(1 - C_i))} \tag{1}$$

These categories correspond to the definition of sea ice that influences iceberg drift besides other influences (of e.g. water velocity and wind) (light), or that solely steers iceberg drift (heavy) (Herrmannsdörfer et al., 2024). Note that the sum of light and heavy sea ice ($CI > 15\%$) reflects all sea ice conditions, relevant for iceberg simulations. As $CI = 15\%$ is commonly defined as sea ice edge, the analysis of sea ice with $CI > 15\%$ may also be used in other geophysical purposes.

We analyse the occurrence of light and heavy sea ice in Topaz and Barents-2.5 in the Fig.9, 10 and 11. The average and maximum spatial extent of light sea ice (solid and dashed line in Fig.9) and heavy sea ice (point and line hatches in Fig.9) within the Barents Sea is larger in Barents-2.5 and reaches further south, compared to the respective extent in Topaz. For the evaluation of temporal variability in the representation of the sea ice categories in Topaz and Barents-2.5, we visualise the number of grid cells that contain these sea ice categories, relative to the total number of grid cells in Fig.11. Figure 11 shows that the relative number of grid cells with $CI > 15\%$ and heavy sea ice is larger in Barents-2.5, compared to Topaz, during the years 2010-2014 (see red shading, solid and dashed line). The percentage of light sea ice is larger Topaz (compared to Barents-2.5) in parts of the year (autumn-spring), when a large percentage of the study region is covered by heavy sea ice. From 2020 to 2022, the differences between Topaz and Barents-2.5 are smaller and Topaz has larger sea ice occurrence in some sea ice categories and seasons. As such, in the winter of 2020/2021 Barents-2.5 shows a larger relative number sea ice with $CI > 15\%$ and light sea ice, but smaller occurrence of heavy sea ice, compared to Topaz. Further analysis reveals $-7\%$, $+0.26\%$ and $-6\%$ larger relative number of grid cells with $CI > 15\%$, light and heavy sea ice in Barents-2.5, averaged over the years 2010-2014





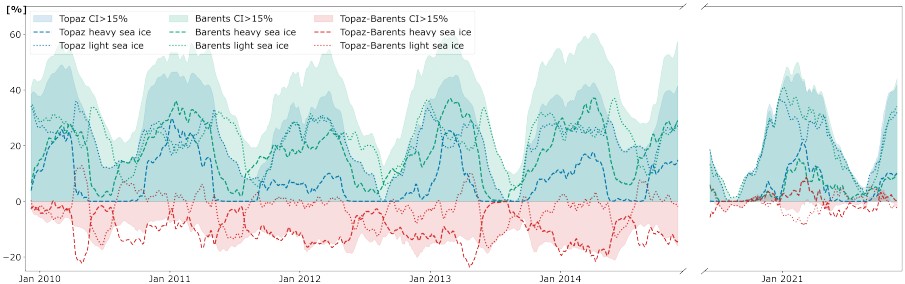

**Figure 11.** Time series of the percentage of grid cells [%] that contain sea ice within the marginal ice zone ($CI > 15\%$, shading), light sea ice ($CI > 15\%$, but $h_{si} < h_{min}$ for $CI >= 90\%$, dotted line) and heavy sea ice ($CI >= 90\%$, $h_{si} > h_{min}$, dashed line) in Topaz (blue) and Barents-2.5 (green) and their difference in Topaz and Barents-2.5 (red). The analysis time period is 2010-2014 and 2020-2022. The comparison is performed on a projected horizontal resolution of $2.5\,\mathrm{km}$, a resampled temporal resolution of $2\,\mathrm{hours}$ (as described in Sect.3.1.1) and a $10\,\mathrm{days}$ rolling average is applied.

and 2020-2022 (Table 2). Larger sea ice extent in Barents-2.5 mostly derives from the hindcast (2010-2014) ($-13\%$, $-2\%$, $-10\%$), while the forecast shows slightly smaller extent ($+9\%$, $+5\%$, $+4\%$) than Topaz.

*Sea ice drift*

Figures 12 and 13 show the results of the comparison between the sea ice velocity variable in Topaz and Barents-2.5. Spatial and
temporal differences exist. Largest spatial differences are visible in regions with average $CI \leq 15\%$, especially south(-west) of Svalbard (Fig.12). In this regions, and the largest parts of the study region, the sea ice drift velocity is larger in Barents-2.5, compared to Topaz. Topaz views slightly larger drift speeds in the Eurasian Basin. Spatial drift speed differences are small in the northern part of the Barents Sea. The sea ice speed difference in Topaz and Barents-2.5 has an annual cycle (Fig.13), that also shows in the spatial drift speed distribution (Fig.12). Largest spatial average speeds, model difference and variability in
the study region is visible from autumn to spring (Fig.13). Smallest spatial average speeds, model difference and variability in the study region is visible in summer. The model differences are in the same order of magnitude than the absolute speed values. Further analysis revealed a mean sea ice drift speed difference of $-0.068\,\mathrm{m\,s^{-1}}$ between Topaz and Barents-2.5, averaged over the Barents Sea and the years 2010-2014 and 2020-2021 (Table 2). The difference between the years of 2010-2014 and 2020-2021 is small.


### 3.1.3   Atmospheric variables

*10m wind*

Figure 14 and 15 show the difference of the 10m wind velocity in ERA5 and CARRA. The 10m wind speed difference between ERA5 and CARRA varies spatially in the Barents Sea (Fig.14). The largest differences are in the northern part of the domain,
where ERA5 provides larger wind speeds than CARRA. CARRA has larger wind speeds in the southern parts of the domain.



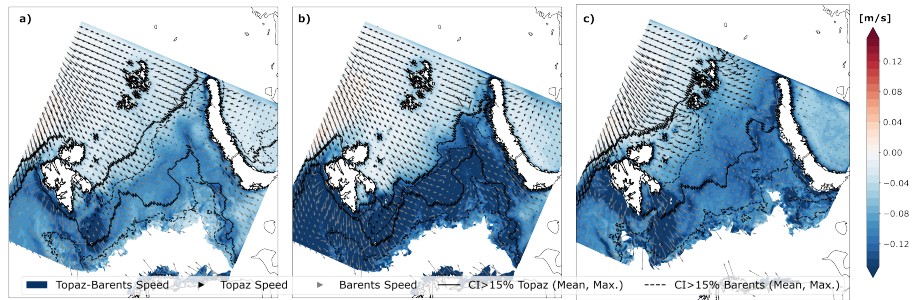

**Figure 12.** Map of sea ice drift speed difference (Topaz-Barents2.5, colours) and sea ice drift directions (arrows) and regions with (average and max.) sea ice concentration $CI > 15\%$ in Topaz (solid line) and Barents (dashed line) in the Barents Sea averaged over a) the analysis time period 2010-2014 and 2020-2022, b) winter and spring month (Dec-May) and c) summer and autumn (Jun-Nov) of these years. The comparison is performed on a horizontal resolution of $2.5\,\mathrm{km}$.

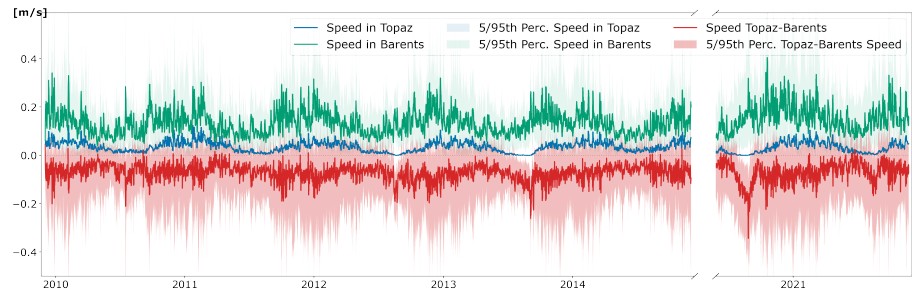

**Figure 13.** Time series of spatial average sea ice drift speed in Topaz and Barents-2.5 (blue, green) and their difference (Topaz-Barents2.5, red) in the Barents Sea for the analysis years 2010-2014 and 2020-2022. The shaded areas show the variability of speeds and the difference in Topaz and Barents-2.5 in the domain, excluding the extreme values (5th and 95th Percentile). The comparison is performed with a temporal resolution of $2\,\mathrm{hours}$ on the temporally resampled datasets (Sect.3.1.1). Data points without sea ice are excluded from this plot.

The spatial model differences are most pronounced in winter to spring and smallest in summer to autumn (Fig.14 and Fig.15, shading). Despite the spatial differences, the spatial wind speed difference mean (and its variability in the domain) is relatively small compared to the absolute wind speeds, especially in spring and summer, and slightly larger from autumn to late winter (Fig.15). An annual cycle of spatial mean wind speed exposes slightly larger wind speeds in ERA5 in the first half of the year and slightly larger wind speeds in CARRA in the second half of the year. However, the short-term variability of the wind speed is partly as large as the seasonal cycle. There is also large variability between the years. Further analysis yielded a wind speed difference of $+0.0007\,\mathrm{m\,s^{-1}}$ between ERA5 and CARRA, averaged over the study region and study years (Table 2).



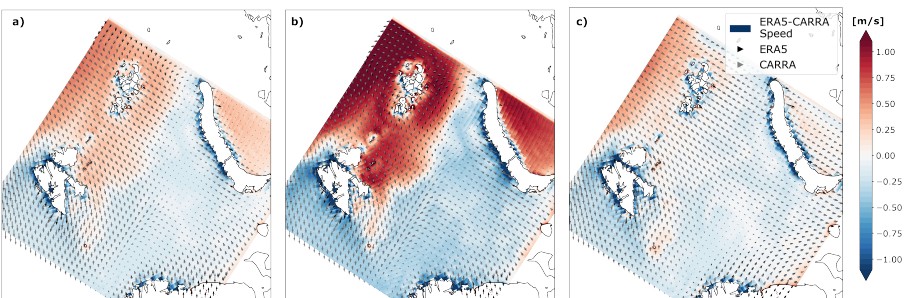

**Figure 14.** Map of 10 m wind speed difference (ERA5-CARRA, colours) and direction (arrows) in the Barents Sea for averaged over a) the analysis time period 2010-2014 and 2020-2022, b) winter and spring month (Dec-May) and c) summer and autumn (Jun-Nov) of these years. The comparison is performed on a horizontal resolution of 2.5 km.

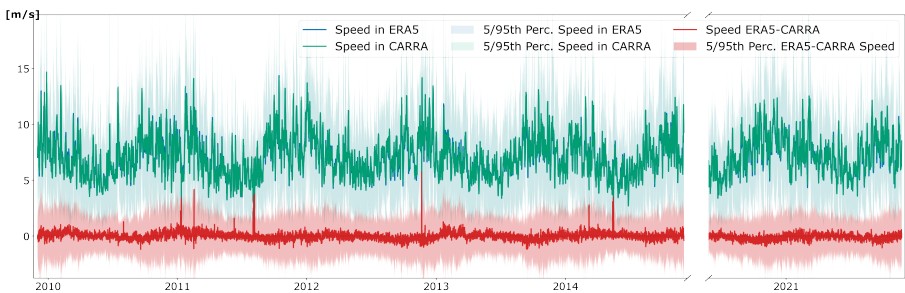

**Figure 15.** Time series of spatial average 10m wind speed in Topaz and Barents-2.5 (blue, green) and their difference (Topaz-Barents-2,5, red) in the Barents Sea for the analysis years 2010-2014 and 2020-2022. The shaded areas show the variability of speeds and the difference in Topaz and Barents-2.5 in the domain, excluding the extreme values (5th and 95th Percentile). The comparison is performed with a temporal resolution of 2 hours on the temporally resampled datasets (Sect.3.1.1).

## 3.2 Statistical analysis of selected model variables along the main iceberg pathways in the Barents Sea

In the previous Sect., we statistically compared selected variables from the models Topaz, Barents-2.5, ERA5 and CARRA in the domain of the Barents Sea and the years 2010-2014, 2020-2021. These variables are used in the sequel paper (Herrmanns-dörfer et al., 2024) as input (forcing) data for our iceberg drift and deterioration model. There, icebergs are seeded in the period of interest at five major sources in the Barents Sea and the drift and deterioration of these icebergs under the influence of wind, waves, water velocity, air and water temperatures are simulated. This Sect. analyses the atmosphere, ocean and sea ice variables, along the simulated trajectories in (Herrmannsdörfer et al., 2024).

### 3.2.1 Data pre-processing

The iceberg model used in Herrmannsdörfer et al. (2024) assimilates environmental data close to the momentary iceberg position at 2 − hourly time steps, as shown in Fig.16, to update iceberg position and size. This step is repeated along one iceberg





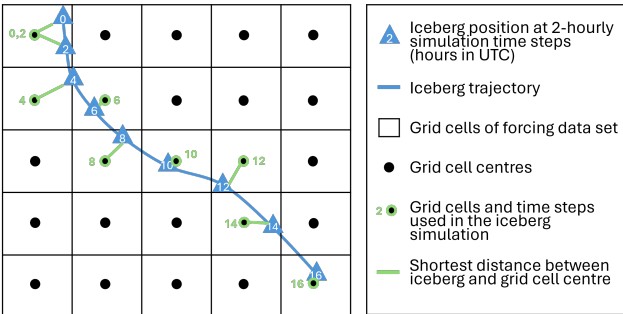

**Figure 16.** Assimilation of environmental data into the model for iceberg drift and deterioration and resulting subset of the environmental data in the spatial and temporal domain. For every $2 - \mathrm{hourly}$ time step (numbers), the nearest grid cell (grid) is estimated by the distance (green line) between iceberg (blue triangle) and grid cell centre (dots). The iceberg is then forced by the data from the respective time step and nearest grid cell. Note the resulting subset of the environmental data (green circles) along the simulated iceberg trajectory (blue line).

trajectory, and then for a large number of iceberg trajectories. The atmosphere, ocean and sea ice variables, that contributed
to the simulations of iceberg drift and deterioration are a temporal and spatial subset of the environmental data sets presented
in Sect. 3.1. The subset represents environmental conditions in the seasons and regions of the Barents Sea, when and where
icebergs occur in the iceberg simulations. In the following, we call those regions and seasons *iceberg pathways*, and the tempo-
ral and spatial subset of the environmental data *data points*. The spatial occurrence of icebergs in the Barents Sea is shown in
Fig. 17. As the iceberg simulation reflect the conditions in the Barents Sea as closely as possible, icebergs may occur in these
pathways. Note, that the pathways vary slightly for the different environmental models, as they cause different trajectories
in the iceberg simulations (Herrmannsdörfer et al., 2024). In this Sect., we provide statistics of the variable deviations in the
named environmental models in the iceberg pathways of the Barents Sea.

Further, we differentiate how many individual iceberg trajectories and data points (in the iceberg pathways) contain specific
values (val, e.g. $CI > 15\%$) by

$$p_{\mathrm{traj}}(val) = \frac{\sum_{i=0}^{I} val(i)}{I} [\%] \qquad (2)$$

$$p_{\mathrm{dp}}(val) = \frac{\sum_{dp=0}^{D} val(dp)}{D} [\%] \qquad (3)$$

where $I$ is the total number of individual iceberg trajectories $i$, $D$ is the total number of individual data points $dp$ and $val$ is the
occurrence (logical statement returning 1 for true and 0 for false) of these values in trajectories (traj.) or data points (dp).





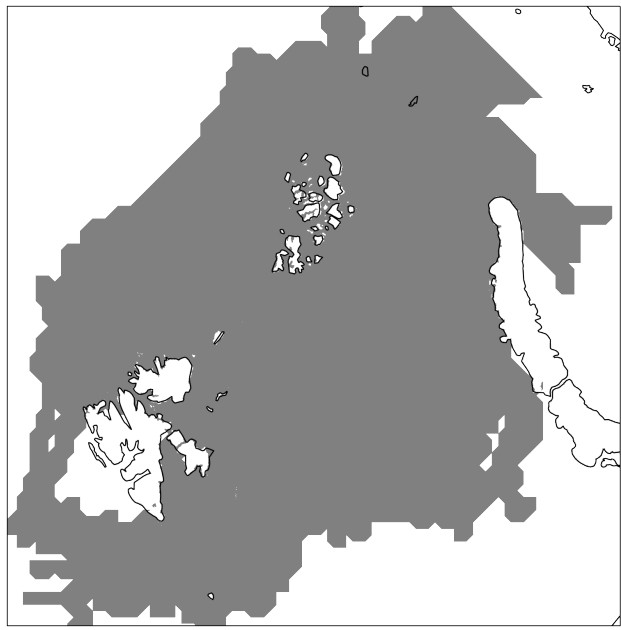

**Figure 17.** Spatial occurrence of icebergs in the Barents Sea, based on the simulations in Herrmannsdörfer et al. (2024). Thereby, grey colouring marks regions with iceberg occurrence.

### 3.2.2 Ocean and sea ice variables

#### *Sea surface temperature*

In the main iceberg pathways in the Barents Sea, including the data points from the described spatial and temporal subset, the $SST$ is predominantly negative, as $78\%$ of data points from Topaz and $89\%$ from Barents-2.5 show $SST$ below zero (Table 3).

The $SST$ minimum is $-1.9°\mathrm{C}$ in Topaz and $-1.88°\mathrm{C}$ in Barents-2.5. The $SST$ average over the spatial and temporal subset is $-0.86°\mathrm{C}$ in Topaz and $-1.26°\mathrm{C}$ in Barents-2.5. Thus the mean difference of $SST$ in Topaz and Barents for the iceberg pathways is $-0.41°\mathrm{C}$.

#### *Sea surface velocity*

In the iceberg pathways, the sea water speed is in average $0.06\,\mathrm{m\,s^{-1}}$ (Topaz) and $0.09\,\mathrm{m\,s^{-1}}$ (Barents-2.5), and top speeds are $0.72\,\mathrm{m\,s^{-1}}$ and $2.00\,\mathrm{m\,s^{-1}}$, respectively (Table 3). Thus, Topaz has smaller average ($-0.034\,\mathrm{m\,s^{-1}}$ ) and top sea water speed ($-1.29\,\mathrm{m\,s^{-1}}$ ). Mean speeds and their model-difference ($\Delta v_\mathrm{w} = -0.05\mathrm{m\,s^{-1}}$) are slightly larger when only accounting for conditions relevant for iceberg drag by ocean velocity (without heavy sea ice).

#### *Sea ice concentration*

Sea ice ($CI > 0\%$) is present in $57\%$ of the Topaz- and $99\%$ of the Barents-2.5-forced ($\Delta p_\mathrm{traj(Topaz-Barents2.5)} = -42\%$)





**Table 3.** Statistics of ocean, sea ice and atmosphere variables in Topaz and Barents-2.5, ERA5 and CARRA for the main iceberg pathways in the Barents Sea. The variables are aggregated over the spatial grid cells and time steps, used in the iceberg simulations (Sect. 3.2.1, Herrmannsdörfer et al. (2024)). The variables are sea surface temperature ($SST$), sea surface speed ($v_{\mathrm{w}}$), sea ice concentration ($CI$), sea ice thickness ($h_{\mathrm{si}}$), sea ice drift speed ($v_{\mathrm{si}}$) and 10m wind speed ($v_{\mathrm{a}}$). $p_{\mathrm{dp}}$ and $p_{\mathrm{traj}}$ give the relative number of data points and trajectories, that exhibit specific characteristics, e.g. $SST$ below zero. Some values are given for specific conditions (e.g. $h_{\mathrm{si}}$ in $CI > 0\%$ or light sea ice) or excluding specific conditions that do not contribute to the iceberg drift and deterioration (e.g. $v_{\mathrm{w}}$ without heavy sea ice).

| Variable | Topaz | Barents2.5 | $\Delta(Topaz - Barents2.5)$ |
|---|---|---|---|
| $\varnothing SST\,[^\circ\mathrm{C}]$ | -0.85 | -1.26 | -0.41 |
| $min(SST)\,[^\circ\mathrm{C}]$ | -1.92 | -1.88 | +0.002 |
| $p_{dp}(SST < 0^\circ\mathrm{C})\,[\%]$ | 78 | 89 | -11 |
| $\varnothing v_{\mathrm{w}}\,[\mathrm{m\,s^{-1}}]$ | 0.06 | 0.09 | -0.03 |
| $\varnothing v_{\mathrm{w}}$ (no heavy sea ice) $[\mathrm{m\,s^{-1}}]$ | 0.06 | 0.10 | -0.05 |
| $max(v_{\mathrm{w}})\,[\mathrm{m\,s^{-1}}]$ | 0.72 | 2.00 | -1.29 |
| $\varnothing CI\,(CI > 0\%)\,[\%]$ | 86 | 72 | -5 |
| $\varnothing h_{\mathrm{si}}\,(CI > 0\%)\,[\mathrm{m}]$ | 0.45 | 1.13 | -0.69 |
| $p_{\mathrm{traj}}\,(CI > 0\%)\,[\%]$ | 57 | 99 | -42 |
| $p_{\mathrm{dp}}\,(CI > 0\%)\,[\%]$ | 68 | 95 | -23 |
| $\varnothing CI\,(CI > 15\%)\,[\%]$ | 86 | 90 | -4 |
| $\varnothing h_{\mathrm{si}}\,(CI > 15\%)\,[\mathrm{m}]$ | 0.57 | 1.41 | -0.84 |
| $p_{\mathrm{traj}}\,(CI > 15\%)\,[\%]$ | 37 | 69 | -32 |
| $p_{\mathrm{dp}}\,(CI > 15\%)\,[\%]$ | 53 | 76 | -23 |
| $\varnothing CI$ (light sea ice) $[\%]$ | 84 | 83 | +0.01 |
| $\varnothing h_{\mathrm{si}}$ (light sea ice) $[\mathrm{m}]$ | 0.45 | 0.95 | -0.5 |
| $p_{\mathrm{traj}}$ (light sea ice) $[\%]$ | 37 | 68 | -31 |
| $p_{\mathrm{dp}}$ (light sea ice) $[\%]$ | 46 | 44 | +4 |
| $\varnothing CI$ (heavy sea ice) $[\%]$ | 1.38 | 2.03 | -0.66 |
| $p_{\mathrm{traj}}$ (heavy sea ice) $[\%]$ | 8 | 43 | -35 |
| $p_{\mathrm{dp}}$ (heavy sea ice) $[\%]$ | 7 | 32 | -25 |
| $p_{\mathrm{dp}}\,(CI \geq 90\%)\,[\%]$ | 36 | 56 | -20 |
| $p_{\mathrm{dp}}(CI \geq 90\%, h_{si} > h_{min})\,[\%]$ | 19 | 58 | -39 |
| $\varnothing v_{\mathrm{si}}\,(CI > 15\%)\,[\mathrm{m\,s^{-1}}]$ | 0.09 | 0.12 | -0.02 |
| $\varnothing v_{\mathrm{si}}$(light sea ice) $[\mathrm{m\,s^{-1}}]$ | 0.10 | 0.13 | -0.04 |
| $\varnothing v_{\mathrm{si}}$(heavy sea ice) $[\mathrm{m\,s^{-1}}]$ | 0.07 | 0.09 | -0.02 |
| $max(v_{\mathrm{si}}\,(CI > 15\%)\,[\mathrm{m\,s^{-1}}]$ | 0.59 | 1.57 | -1.0 |
| Variable | ERA5, CARRA | Topaz, Barents2.5 | $\Delta(ERA5 - CARRA), \Delta(Topaz - Barents2.5)$ |
| $\varnothing v_{\mathrm{a}}\,[\mathrm{m\,s^{-1}}]$ (no heavy sea ice) | 6.69, 6.60 | 6.61, 6.68 | +0.08, -0.08 |
| $max(v_{\mathrm{a}})\,[\mathrm{m\,s^{-1}}]$ (no heavy sea ice) | 24.74, 26.17 | 25.80, 26.17 | -1.43, -0.37 |





iceberg trajectories (Table 3). It is present in 68 and 95% ($\Delta p_{\mathrm{dp(Topaz-Barents2.5)}} = -23\%$) of the data points. In the presence of sea ice ($CI > 0\%$), the mean ice concentration is 67% (Topaz) and 72% (Barents-2.5) ($\Delta CI_{\mathrm{(Topaz-Barents2.5)}} = -5\%$).

Sea ice is most relevant for the iceberg deterioration and drift for sea ice concentrations greater 15%. Sea ice with concentration greater 15% is present in 37% and 69% of the Topaz and Barents2.5-forced trajectories and 53% and 76% of the data points (Table 3). Light sea ice conditions are present in 37% and 68% of the trajectories and 46% and 44% of the data points. Heavy sea ice conditions are present in 8% and 43% of the trajectories and 7% and 32% of the data points. In sea ice conditions relevant for iceberg drift and deterioration ($CI > 15\%$), the mean sea ice concentrations are 86% and 90%. In evaluation of

heavy sea ice conditions, 36% (Topaz) and 56% (Barents-2.5, $\Delta p(CI \leq 90\%)_{\mathrm{dp,(Topaz-Barents2.5)}} = 20\%$) of the data points have a $CI$ of at least 90%.

*Sea ice thickness*

The average sea ice thickness is $0.45\,\mathrm{m}$ (Topaz) and $1.13\,\mathrm{m}$ (Barents-2.5), in the presence of sea ice ($CI > 0$) (Table 3).

In those conditions, the average $h_{\mathrm{si}}$ difference in Topaz and Barents-2.5 is $-0.69\,\mathrm{m}$. Sea ice thickness is relevant for the iceberg drift in light sea ice conditions and is needed to identify heavy sea ice conditions. The average sea ice thickness is $0.45\,\mathrm{m}$ (Topaz) and $0.95\,\mathrm{m}$ (Barents-2.5, $\Delta h_{\mathrm{si,(Topaz-Barents2.5)}} = 0.5\mathrm{m}$) in light sea ice conditions and $1.38\,\mathrm{m}$ (Topaz) and $2.03\,\mathrm{m}$ (Barents-1.5, $\Delta h_{\mathrm{si,(Topaz-Barents2.5)}} = -0.66\mathrm{m}$) in heavy sea ice conditions. In the evaluation of heavy sea ice, the sea ice thickness is fulfilling the strength criterion $h_{\min}$ (Eq. 1) in 19% (Topaz) and 58% of the data points (Barents-2.5,

$\Delta p_{dp}(CI \geq 90\%, h_{si} > h_{min}) - 39\%$) with sea ice of concentration of at least 90%.

*Sea ice drift*

In the iceberg pathways of the Barents Sea, sea ice drift speeds are in average $0.10\,\mathrm{m\,s^{-1}}$ (Topaz) and $0.13\,\mathrm{m\,s^{-1}}$ (Barents-2.5) in light sea ice conditions and slightly smaller in heavy sea ice conditions with $0.07\,\mathrm{m\,s^{-1}}$ (Topaz) and $0.09\,\mathrm{m\,s^{-1}}$ (Barents-

2.5) (see Table 3). Top sea ice speeds are $0.59\,\mathrm{m\,s^{-1}}$ (Topaz) and $1.57\,\mathrm{m\,s^{-1}}$ (Barents-2.5) and occur in light sea ice conditions Thus, Barents has $0.02\,\mathrm{m\,s^{-1}}$ larger average and $0.99$ larger maximum sea ice drift velocities contributing to the iceberg drift.

### 3.2.3  Atmospheric variables

*10m wind*

For the iceberg pathways of the Barents Sea, the 10m wind speed is in average $6.6\,\mathrm{m\,s^{-1}}$ and maximum $26.2\,\mathrm{m\,s^{-1}}$ (Table

3). Thereby, ERA5 has $0.08\,\mathrm{m\,s^{-1}}$ larger average and $1.43\,\mathrm{m\,s^{-1}}$ smaller top wind speeds, compared to CARRA. Iceberg pathways that are forced by Topaz, have $0.08\,\mathrm{m\,s^{-1}}$ lower average and $0.37\,\mathrm{m\,s^{-1}}$ lower maximum wind speed, compared to pathways of that were forced by Barents-2.5.



## 4   Discussion

In the following, the differences between the selected variables in Topaz and Barents-2.5, as well as, ERA5 and CARRA are
discussed for the domain of the Barents Sea and the years 2010-2014 and 2020-2021, and set in relation to the knowledge from
literature. Note that the comparisons and quality assessments from literature have limited applicability for this study, as they
are referring to partly different products (e.g. Barents-2.5 forecast EPS), spatial (e.g. the entire Arctic or a point observation)
and temporal scales (e.g. different years), and they are partly based on sparse observations with potentially large observational
and representative error.


The literature showed a number of differences and similarities of the ocean, sea ice and atmospheric models of this study,
Topaz, Barents-2.5, ERA5 and CARRA. Similarities are for example the physical simulation of the variables of interest as
prognostic variables in all four models a similar model setup op Topaz and Barents-2.5 (MDS, 2023; Duarte et al., 2022;
Hersbach et al., 2020; Copernicus Climate Change Service (C3S), 2023). Further, the models are using output of the others
models (or closely related versions) as initial conditions, forcing at the surface and lateral boundaries (Sect. 2). This may cause
similarities in the model outputs but may also pass on systematic errors. A more detailed discussion of the model similarities
and differences is provided in the Appendix (Sect. A).

### 4.1   Sea surface temperature

The sea surface temperature ($SST$) is a prognostic variable of the ocean component in both Topaz and Barents-2.5, and con-
straint to satellite-based observations (MDS, 2023; Hackett et al., 2022; Röhrs et al., 2023). This study revealed differences in
the model outputs over the Barents Sea and the years 2010-2014, 2020-2021.
We find average $0.55°C$ lower $SST$ in the Barents-2.5 model, that coincide with more extensive sea ice of the same model
(Sect. 3.1). The spatial and temporal $SST$ differences between Topaz and Barents-2.5 follow the bathymetry, ocean velocities
and sea ice characteristics.


Largest spatial model differences of the $SST$ (larger in Topaz) are present along the northward inflow of warm Atlantic wa-
ter along the west-coast of Svalbard (e.g. the West-Spitsbergen-Current (WSC) and Fram Strait Branch of the Atlantic Water
Boundary Current in the Arctic Ocean (FSAW)), as well as, inflow into the Storfjorden Trough, Bjørnøya Trough and Hopen
Trench (Fig.5). These deviations between Topaz and Barents-2.5 may derive from present model errors, as Topaz has a large
positive $SST$ bias in those specific regions (Xie et al., 2017). The bias in Topaz is (at least partially) resolving from problems
in simulating the circulation of Atlantic water inflow and the topographic steering (Xie et al., 2017). A negative $SST$ bias and
large $SST$ mismatches in the marginal ice zone are found in Barents-2.5 (Röhrs et al., 2023). Further, the $SST$ deviations
in Topaz and Barents-2.5 forecast might derive from different assimilated observations and the high observational error of
some assimilated $SST$ observations in those regions (Xie et al., 2017). $SST$ differences in Topaz and Barents-2.5 may also
derive from the representation of the sea ice (Sect. 3.1), as the $SST$ is consistent with the sea ice representation, due to the



2-way-coupling of the model ocean and sea ice components (Duarte et al., 2022). The described regions of large $SST$ model deviation also outline the approximate maximum sea ice edge in the Barents Sea and also large $CI$ deviations (see Fig. 10). In contrast, Barents-2.5 has slightly higher $SST$ in the northern part of the domain (Fig. 5), as it defines slightly larger $SST$ under sea ice ($-1.88°$C), compared to Topaz ($-1.92°$C, Table 3). The large $SST$ bias along the shelf edge north of Svalbard

(Xie et al., 2017) does not cause larger deviations between the two models.

    The largest model differences of $SST$ in summer (larger $SST$ in Topaz) (Fig. 6) reflect the known warm summer bias in Barents-2.5 forecast and Topaz (Xie et al., 2017; Xie and Bertino, 2022; Röhrs et al., 2023). It also coincides with previously observed too fast sea ice retreat in Topaz (Xie et al., 2017; Xie and Bertino, 2022) and a delayed sea ice retreat in Barents.2-5

in this study (Sect. 3.1). Smaller model differences of $SST$ in this study in winter and spring (larger $SST$ in Barents-2.5) (Fig. 6) reflect the previously found smaller cold bias of Topaz and Barents-2.5 hindcast in winter (Xie et al., 2017; Xie and Bertino, 2022; Idžanović et al., 2024). The decreased model difference may derive from compensating smaller Topaz $SST$ under sea ice and warmer Topaz $SST$ outside the sea ice (Fig. 5). Thereby, Topaz is closer to the observations than the Barents-2.5 hindcast.

### 4.2    Sea ice concentration and thickness

Sea ice concentration and thickness are prognostic variables of the sea ice component in Topaz and Barents-2.5 (MDS, 2023; Duarte et al., 2022; Röhrs et al., 2023). Topaz and Barents-2.5 forecast assimilate satellite-based information of $CI$, however from different sources and at different resolutions. Topaz assimilates satellite products of the sea ice thickness from late October to early April, in addition (MDS, 2023; Hackett et al., 2022; Röhrs et al., 2023). The Barents-2.5 hindcast does not assimilate information after the initialisation in the domain, but uses Topaz at the lateral boundaries (Idžanović et al., 2024).


    In this study, we analyse the sea ice thickness and concentration, as well as, the occurrence of the sea ice categories, light ($CI > 15\%$, excluding heavy sea ice) and heavy sea ice ($CI \geq 90\%$, $h_{\mathrm{si}} > h_{\mathrm{min}}$), that relate to the sea ice contributing to the iceberg drift and deterioration (Sect. 3.2.1). We find that Barents-2.5 has in general more extensive sea ice with average larger $CI$ (10%), $h_{\mathrm{si}}$ (0.23 m) and more grid cells with sea ice in the domain (7% for $CI > 15\%$ and 6% for heavy sea ice, Sect. 3.1).

This may be explained by known underestimation of $CI$, sea ice area with $CI > 15\%$ and $h_{\mathrm{si}}$ in Topaz (Xie et al., 2017; Xie and Bertino, 2022) and general overestimation in Barents-2.5 (Röhrs et al., 2023). Thereby Topaz is closer to the observations as the Barents-2.5 hindcast. Topaz has largest $CI$ biases along the sea ice edge (Xie et al., 2017). The $CI$ in Barents-2.5 is too large throughout most of the domain and time period and is in general described as skilful in periods with enough data assimilation (Röhrs et al., 2023).


    A consequence of the described under- and overestimation is also the larger and more southern mean and maximum spatial extent of $CI > 15\%$ and heavy sea ice within the Barents Sea in the Barents-2.5 model, compared to Topaz (Fig. 9, 10). We find largest spatial model differences of $CI$ and $h_{\mathrm{si}}$ around northern Svalbard (Fig. 9, 10).





The sea ice coverage in the domain is larger in Barents-2.5 during most of the seasonal cycle, however, the area with light and heavy sea ice is larger in Topaz during short time periods (Fig. 11). A larger extent of light sea ice in Topaz during parts of the year may be related with the larger percentage of heavy sea ice Barents-2.5, also resulting in a smaller average model difference ($0.26\%$) of data points with light sea ice. In comparison, quality assessments of Topaz describe an underestimated $CI$ in winter, both under- and overestimated $h_{\mathrm{si}}$ in spring, accurate $CI$ in summer, and underestimated $h_{\mathrm{si}}$ in autumn (Xie and Bertino, 2022). Model differences in the seasonal cycle also indicate a longer melt season in Barents-2.5, an earlier spread of heavy sea ice but a similar freeze-up (for light sea ice), compared to Topaz (Fig. 11). Too fast decline and extent of sea ice in autumn and spring in Topaz in previous studies (Xie et al., 2017; Xie and Bertino, 2022), may explain the model differences in spring and may indicate a similar hastened autumn extent of (light) sea ice in Barents-2.5.

### 4.3  Sea water surface velocity

The sea water velocity is a prognostic variable in ocean component of Topaz and Barents-2.5, yet it is represented differently (MDS, 2023; Duarte et al., 2022; Röhrs et al., 2023). The models incorporate different resolutions and mechanisms, for example, Barents-2.5 accounts for the effect of air pressure and tides in the calculations of water velocity, and represents local water velocities due to its high horizontal and temporal resolution, while Topaz does not (Röhrs et al., 2023; Röhrs et al., 2023). These mechanisms can be important in simulations of drifting objects, such as icebergs. In this study, we use the sea water surface velocity of Topaz and Barents-2.5, referring to the uppermost sea water layer of $0.2$ to $1.2\,\mathrm{m}$ in Barents-2.5 (Röhrs et al., 2023) and from 0 to $1\,\mathrm{m}$ in Topaz (MDS, 2023).

The representation of local conditions and tides might imply that Barents-2.5 may be suitable for the simulation of trajectories of drifting objects, as in the OpenDrift framework (Röhrs et al., 2023; Idžanović et al., 2023). Despite the simulating efforts, studies discovered low predictive skill for surface water speed and direction in Barents-2.5 (Idžanović et al., 2023), due to the chaotic nature of the system, the scarcity of observations and error statistics (Röhrs et al., 2023). Model skills and their representation of individual mechanisms vary over time- and spatial scales (Röhrs et al., 2023). Some skill could be assigned to versions of the Barents-2.5 forecast that are forced by high resolution atmospheric data (AROME-Arctic) and in conditions with mainly wind driven water velocity, when the otherwise uncertain initial conditions are less relevant (Idžanović et al., 2023). The authors did not find validation of the Topaz water velocity, but similar model setup and assimilated data indicate similar problems in the simulation of the variable. It is assumed in general, that ocean models with low skill in surface velocity may still benefit from velocity caused by tides, wind and topography (Röhrs et al., 2023). In addition, lower horizontal resolution ocean models have smaller gradients and lower velocities in general. Thus, Topaz may suffer from lower resolution bathymetry and missing tidal representation.

In this study, the sea surface speed is larger in Barents-2.5, compared to Topaz, throughout the largest parts of the Barents Sea (Fig. 7), the seasonal cycle during the years 2010-2014 and 2020-2021 (Fig. 8), and their average ($0.047\,\mathrm{m\,s^{-1}}$), Table 2). Large model differences of the water speed (larger speeds in Barents-2.5) exist in coastal areas and bathymetric features



with reduced water depth (e.g. Spitsbergen Bank, Central Bank, islands between Svalbard and Franz-Josef-Land, see Fig. 7),
where large speeds occur during the diurnal cycle caused by the tides, that are only represented in Barents-2.5. The different
representations of the coastline are shown in Fig. 3a). In the deep waters of the Eurasian Basin, Topaz has larger (mostly
southward) sea surface speeds than Barents-2.5 (Fig. 7). The differences are most pronounced in summer and autumn, when
the speeds are largest (in spatial average) and the variability in the domain is largest (Fig. 8).

### 4.4   Sea ice drift

The sea ice component in Topaz and Barents simulates the horizontal advection of sea ice concentration and volume, and thus
the sea ice velocity. Topaz assimilates a coarse sea ice drift product from October-April (MDS, 2023; Hackett et al., 2022;
Röhrs et al., 2023).

This study showed an average $-0.068\,\mathrm{m\,s^{-1}}$ smaller sea ice drift speed in Topaz, compared to Barents-2.5 (Sect. 3.1). The
model difference is largest in low sea ice concentrations and small in high sea ice concentrations and large distances to the
marginal ice zone (Fig. 12). Topaz has slightly higher drift speeds towards the northern Eurasian Basin in this analysis (Fig.
12), where large errors are found in literature (Xie and Bertino, 2022).

The model differences are larger in autumn to spring and smaller in summer, coinciding with larger (smaller) speed and
variability in the domain (Fig. 13). In contrast, previous literature stated overestimation of Topaz sea ice drift in winter and
underestimation in summer (Xie and Bertino, 2022). Note that Topaz assimilates satellite-based sea ice drift observations
during this time (MDS, 2023; Hackett et al., 2022; Röhrs et al., 2023). This supports the too large drift speeds in the Barents-
2.5 forecast from literature and may indicate that the error is even larger than in Topaz. However, the sea ice drift in the
Barents-2.5 forecast is also attributed with some skill in the literature (Röhrs et al., 2023).

### 480   4.5   10m wind

ERA5 and CARRA show similarities and difference in the model setup. 10m wind speed is a prognostic variables in both mod-
els (Hersbach et al., 2020; Køltzow et al., 2022). In ERA5, wind speeds are assimilated from a wide spectrum of observation
sources (Hersbach et al., 2020). CARRA assimilates a wide range of variables from ERA5 (Yang et al., 2020b), however the
10m wind is not assimilated (Køltzow et al., 2022). Both atmospheric models, ERA5 and CARRA, assimilate observations of
$CI$ and $SST$, influencing the calculation of wind (Hersbach et al., 2020; Køltzow et al., 2022).

In the literature and this study, wind speeds are very similar in ERA5 and CARRA in general. The resulting average wind
speed difference in the Barents Sea is small (average $+0.0007\,m/s$, Table 2) compared to the absolute wind speeds, however
slightly larger in ERA5. This goes along with previous model comparisons, finding underestimation and larger total error
(model, observational- and representativity- and random error) for ERA5 (for Svalbard) (Giusti, 2024). CARRA was found to



overestimate the wind speed, while still having an added value over ERA5 (Køltzow et al., 2022; Giusti, 2024).

The wind speed difference is specially small over open ocean (Køltzow et al., 2022), due to smaller model errors over ocean and the extensive use of ERA5 output in CARRA (Hersbach et al., 2020; Køltzow et al., 2022; Giusti, 2024). In accordance, this study shows small differences (and larger wind speeds in CARRA) in the southern part of the Barents Sea and in summer (over open water) and largest differences (higher wind speeds in ERA5) in the northern part of the domain and in winter, coinciding with the (largest) extent of the sea ice cover (Fig. 14). This spatial wind speed differences may derive from different representation of sea ice and roughness of ocean and sea ice. Both models assume level sea ice and therefore overestimate near surface winds over multi-year ice (Giusti, 2024), which might occur in the north-western part of the domain. These spatial wind speed differences may also derive from varying $CI$, that both models prescribe from satellite observations (Hersbach et al., 2020; Yang et al., 2020b). Due to improved physical parametrisation and higher resolution satellite observations of sea ice, CARRA was found to have added value over sea ice (Giusti, 2024). Note also that the representation of the surface type, which influences the surface roughness and thus the 10m wind, varies with the horizontal resolution, the topography and land-sea-ice-mask.

The representations of the coastlines in ERA5 and CARRA, that vary with horizontal resolution and topography (Fig. 3a), may also cause the large wind speed difference along the coastlines in Fig. 14. Thereby, CARRA is more accurate over complex topography (Giusti, 2024; Køltzow et al., 2019, 2022).

Seasonal difference between ERA5 and CARRA wind speed in this study may derive from the bipolar spatial difference (Fig. 14 and the seasonal variations of the model errors (Giusti, 2024). Model differences, variability in the domain and model errors are largest in winter and lowest in summer (Fig. 15, Giusti (2024).)

Smaller scale temporal and spatial deviations of ERA5 and CARRA wind speed in this study (Fig. 14, 15) may derive from better representation of temporal and spatial variability in CARRA (Køltzow et al., 2022). However, the sub-grid variability of the wind is large, also for CARRA (Køltzow et al., 2022), especially in complex terrain, e.g. the coastline of Svalbard.

Note that this study analyses the wind over ocean and sea ice, and land grid cells are masked out.

### 4.6 In the iceberg pathways

We analysed model and variable difference in regions and seasons of iceberg occurrence in the Barents Sea, the iceberg pathways (Sect. 3.2.1). The main iceberg pathways cover regions and seasons with partly large model disagreement in Topaz and Barents-2.5. The effect of the model differences on the iceberg simulations will be investigated in (Herrmannsdörfer et al., 2024).





The majority of the simulated icebergs and simulation data points were subject to sea ice, forcing by either Topaz or Barents-2.5. However, Barents-2.5 forces the iceberg simulations with sea ice in $4\%$ to $42\%$ more iceberg trajectories and data points, depending on the sea ice category ($CI > 15\%$, light and heavy sea ice). However, the average $CI$ in the iceberg pathways is similar in Topaz and Barents-2.5, given the category ($\Delta -5\%$ to $+0.001\%$). The average $h_{si}$ difference in Topaz and Barents-2.5 in the iceberg pathways varies between $-0.5\,\mathrm{m}$ and $-0.84\,\mathrm{m}$ depending on the sea ice category, and is thereby larger than

in the analysis of the entire domain. In the evaluation of heavy sea ice, Topaz has less data points with $CI \leq 90\%$, of which even less data points fulfil the strength criterion. We deduct that the frequent occurrence of sea ice within the iceberg pathways, but large model differences in the occurrence and sea ice thickness is influential in the iceberg simulations.

The main iceberg pathways in the Barents Sea also cover regions of large $SST$ difference, where Barents-2.5 provides

$-0.41°\mathrm{C}$ smaller average $SST$ and more data points with negative temperatures to the simulations of iceberg drift and deterioration than Topaz.

Within the iceberg pathways, the difference in water and sea ice speed can be large regionally, however, the average deviation between Barents-2.5 and Topaz is smaller, than in the entire Barents Sea and varies with the sea ice conditions (Table 3). The

deviation of wind speed in ERA5 and CARRA in the iceberg trajectories is small compared to the absolute speed values, but larger than in the entire study domain. The wind speeds vary at the same magnitude for the pathways, determined from different ocean forcing (Topaz and Barents-2.5).

### 4.7    Barents-2.5 hindcast and forecast

We discovered differences between Barents-2.5 hindcast (2010-2014) and forecast (2020-2021), and thus different deviations

compared to Topaz. Although differences may partly be caused by natural multi-year variability, the different data assimilation approaches likely play a large role. As such, larger sea ice extent in the hindcast (compared to the forecast) may be due to multi-year variability and the model state drifting off without data assimilation (Röhrs et al., 2023; Idžanović et al., 2023). Following, the hindcast shows larger sea ice extent, $CI$ and $h_{\mathrm{si}}$ than Topaz (Table 2). In contrast, the Barents-2.5 forecast shows smaller sea ice extent, but similar $CI$ and $h_{\mathrm{si}}$ than Topaz. The deviation between Topaz and forecast is smaller than

between Topaz and the hindcast. For $SST$, the difference between Topaz and Barents-2.5 is slightly larger for the hindcast period, however, the difference between the years of 2010-2014 and 2020-2021 does not reflect the large variations in sea ice. Sea ice and surface water velocities showed no systematic differences between the two time periods and models. Given the significant differences in assimilated data, the differences between 2010-2014 and 2020-2021 are relatively small.

### 5    Conclusions

Although the advantages of using numerical models of atmosphere, ocean and sea ice are various, the discrepancy in the representation of the Arctic domain, reveals the need to evaluate the applicability of these models for the different use-cases. In



this study, ocean, sea ice and atmosphere variables from the models Topaz, Barents-2.5, ERA5 and CARRA are compared statistically in the Barents Sea and over the years 2010-2014 and 2020-2021, and related to knowledge from literature. In the second part of study, (Herrmannsdörfer et al., 2024), the knowledge about the deviations in ocean, sea ice and atmosphere variables is used to study the sensitivity of iceberg drift and deterioration simulations to those varied input variables in the Barents Sea. Moreover, the results of this part of the study may also be used in other geophysical research and applications of the domain.

In the literature of the ocean, sea ice and atmosphere models, Topaz, Barents-2.5, ERA5 and CARRA, we find similarities and differences in the model setup, forcing data, data assimilation approach, resolution and representation of the bathymetry. Further, we find that the models are interconnected by using the respective other models (or different version of a similar setup) as forcing at the ocean, sea ice and atmosphere interface and the lateral boundaries. This may cause similarities in the model output, but also inheritance of errors.

Analysing the variables of interest, we find that Barents-2.5, compared to Topaz, has larger $CI$ and $h_{\text{si}}$, as well as, larger coverage of the Barents Sea with (light and heavy) sea ice. Those differences may be related to the general underestimation of $CI$ and $h_{\text{si}}$ in Topaz and general overestimation in Barents-2.5. Further, the temporal characteristics of sea ice growth and decline in the Barents Sea vary in the sea ice models. Compared to the known too fast decline and freeze-up in Topaz, the melt season is delayed and the sea ice advance is similar in Barents-2.5.

Average lower $SST$ in Barents-2.5, compared to Topaz, are consistent with the more extensive representation of sea ice in the same model. The spatial and temporal differences of $SST$ in Topaz and Barents-2.5 coincide with the sea ice edge, the bathymetry and ocean currents with warm Atlantic water inflow, agreeing with previously found Topaz bias due to simulating the Atlantic inflow and effects of the bathymetry.

Sea water surface speed and sea ice drift speed are in average larger in Barents-2.5, compared to Topaz, and especially along bathymetric features with reduced water depth in the Barents Sea (for sea water speed), and the sea ice edge (for the sea ice drift). The sea surface velocity must be treated especially careful, as the general lack of observations, limits the predictive skill of its forecasts, and limits the constriction to observations in reanalyses and forecasts. However, Barents-2.5 may still benefit from the representation of tides, the effect of air pressure on the water surface and high spatial and temporal resolution, in comparison to Topaz.

10m wind in ERA5 and CARRA is well documented in literature and is very similar over ocean surface. However, we find that CARRA has slightly larger wind speeds over water surfaces and ERA5 has larger speeds over sea ice, which may derive from different representation of surface roughness over water and sea ice, or prescription of different $CI$ products. Further, we find large differences in coastal areas, where representation of the topography and land-sea-mask varies. Those differences

shall be seen in the light of previously attributed added value of CARRA over ERA5 in complex topography and over sea ice. Differences are in general small, compared to the absolute wind speeds.

Generally similar ocean, sea ice and atmosphere variable differences are found along the main iceberg pathways of the Barents Sea. However, for the individual variables, the model differences are partly more/less strongly pronounced. We find, that especially the difference in sea ice representation are relevant, as most simulated iceberg trajectories and simulation time steps encounter sea ice.

We emphasise that this study is limited to the Barents Sea in the years of 2010-2014 and 2020-2021. It is also limited by the availability of detailed description of the atmosphere, ocean and sea ice models, and the availability of quality assessments of the variables. This study may be extended to a larger number of years and variables for the use in other applications.

*Data availability.* Data from ERA5 and CARRA are retrieved from the Copernicus Climate Data Store (Hersbach et al., 2023; Schyberg et al., 2023). The Arctic Ocean Physics Renanalysis (Topaz) is available in Copernicus Marine (MDS, 2023). The Barents-2.5 forecast and 605    hindcast are stored by MET Norway (MET-Norway, a, b). The iceberg pathways in the Barents Sea, from simulations of iceberg drift and deterioration, are described in the subsequent paper Herrmannsdörfer et al. (2024).

## Appendix A: Comparative study of ocean, sea ice and atmospheric models

Two selected models of ocean and sea ice and two atmospheric models are compared regarding their model resolution, model setup, forcing, data assimilation, physical representation and quality of the variables of interests, based on literature. This shall 610    help to understand the model differences and to estimate the impact of those differences on e.g. simulations of iceberg drift and deterioration.

### A1    Comparative study of Topaz reanalysis, Barents-2.5 hindcast and forecast

#### A1.1    Model setup

Topaz and Barents-2.5 are composed of the Hybrid Coordinate Ocean Model (HYCOM) that couples the ocean (Regional 615    Ocean Modeling System, ROMS) and sea ice component (Los Alamos Sea Ice Model, CICE) (Duarte et al., 2022). The described version of the sea ice model is resolving the thermodynamic processes and elastic-viscous-plastic (EVP) rheology (Sakov et al., 2012; MDS, 2023; Hackett et al., 2022).

#### A1.2    Horizontal and temporal resolution

The horizontal resolution is $12\,\mathrm{km}$ for Topaz, $2.5\,\mathrm{km}$ for Barents-2.5, and widely homogeneous in the domain (MDS, 2023; 620    Röhrs et al., 2023) (Table 1). Both model grids have horizontal curvilinear coordinates. The models use topography and





bathymetry with partly different resolution and levels of detail. For example, Barents-2.5 defines the land-sea-mask by a minimum water depth of 10m (Röhrs et al., 2023). Barents-2.5 benefits from a high horizontal resolution of the variables, bathymetry and coastlines. Topaz is limited in its representation of small scale processes and local gradients due to its coarse resolution. Note that the representation of vertical gradients, velocities and their resolution is disregarded in this study.

The temporal resolution is daily to monthly in Topaz and hourly in Barents-2.5 (MDS, 2023; Röhrs et al., 2023). Contrary to Topaz, the high temporal resolution of Barents-2.5 allows the representation of diurnal and fast processes, e.g. tides, sea surface elevation influenced by air pressure.

### A1.3 Forcing and boundary conditions

Topaz is forced at the atmosphere-ocean and atmosphere-sea ice interface by $6 - \mathrm{hourly}$ atmospheric fields of e.g. air pressure, wind and temperature from $0.25°$ ERA5 (Hackett et al., 2022) (Fig. 1). The Barents-2.5 forecast is forced at the atmosphere-ocean and atmosphere-sea ice interface by values of e.g. air temperature, wind, humidity, precipitation and cloud cover. The used version of Barents-2.5 in 2020 and 2021 is forced by the regional arctic atmospheric forecast by Met-Norway (AROME Arctic) on a grid that is very similar to the one of Barents-2.5 (Röhrs et al., 2023; Duarte et al., 2022) (Fig. 1). The updated

version of Barents-2.5 (2022-present, not used) is run as 24-member ensemble (EPS), in which 4 members of the Barents ensemble are forced by AROME Arctic. 20 members of the Barents-ensemble are forced by an ensemble of the global atmospheric forecast by ECMWF (ECMWF ENS) with a resolution of $10\,\mathrm{km}$. Validation showed higher skill for some variables (e.g. sea water surface speed) forced by higher resolution data (AROME Arctic) at the ocean, sea ice, atmosphere interface of the Barents-2.5 forecast (Idžanović et al., 2023). Following lower skill is expected for Barents-2.5 forecast variables forced

by lower resolution ECMWF ENS, and potentially Topaz variables forced by the related reanalysis ERA5. The Barents-2.5 hindcast is forced by atmospheric information from ECMWF ENS (Idžanović et al., 2024).

The Barents-2.5 forecast is forced at the lateral boundaries by a daily version of the Topaz4 forecast (e.g. temperature, salinity, sea surface elevation, ocean water velocities) (Röhrs et al., 2023; Duarte et al., 2022). The Barents-2.5 hindcast is

645 forced at the lateral boundaries by the Topaz reanalysis (Idžanović et al., 2024). This approach might cause inheritance of errors in Barents-2.5. However, some missing effects in Topaz4 (e.g. tides) are corrected during the assimilation into Barents-2.5.

### A1.4 Data assimilation

Topaz assimilates observations of e.g. sea ice concentration, thickness and drift velocities, sea (surface and profile) temperature

and sea level anomaly (Fig. 1). The $SST$ is assimilated from the OSTIA system (Operational Sea Surface Temperature and Ice Analysis, https://doi.org/10.48670/moi-00165) by the UK's Met Office, combining different satellite products, in-situ ship and buoy observations and the assimilated temperature profiles from research cruises. The assimilated sea ice information originate from satellite observations, in detail, the $25\,\mathrm{km}$ OSI-SAF sea ice concentration, the combined SMOS/CryoSAT2




(CS2SMOS) ice thickness (available from late October to early April only) and the $35\,\mathrm{km}$ sea ice drift (October-April, not valid for marginal ice zone) (MDS, 2023; Hackett et al., 2022). The Barents-2.5 forecast assimilates observations of $SST$ from bias- and consistency-corrected satellite products (e.g. AVHRR, VIIRS, SLSTR Sentinel-3), $CI$ from satellite-based microwave brightness temperature observations at different resolutions and frequencies (AMSR2,GCOM-W1) that in combination have close to model resolution (Röhrs et al., 2023) (Fig. 1). Ocean velocities are neither assimilated nor perturbed and, thus, develop by wind forcing and $SST$ (Röhrs et al., 2023; Idžanović et al., 2023). The Barents-2.5 hindcast is a free-run, which is re-initiated daily without assimilation of ocean and sea ice information after the initialisation in the domain (Idžanović et al., 2024).

### A1.5 Prognostic variables

Prognostic variables are (amongst others) sea water temperature and velocities (in ocean component ROMS) $h_{\mathrm{si}}$ and $CI$ (in sea ice component CICE). CICE also simulates the horizontal advection of sea ice concentration and volume (Duarte et al., 2022). The surface velocity is steered by e.g wind, waves, tides, and internal effect of turbulence, pressure and density effects (Röhrs et al., 2023). Different definitions of surface water velocity refer to different mechanisms, water depths, time and spatial scales and so do the models and observations. Different applications, such as gravity-based and floating structures, ship routes, rescue operations and tracing of matter, may profit from different definitions. For the simulation of drifting objects, Röhrs et al. (2023) suggests the effective drift velocity including all mechanisms that are relevant for the specific application, integrated over and dependent on the object depth. The representation of the uppermost water layer may differ in Barents-2.5 and Topaz.

### A1.6 Quality assessment of the variables of interest

The following paragraphs summarise the comparison of ocean, sea ice and atmosphere variables to observation in the literature. The quality assessment of Topaz, amongst others, provides systematic errors over the full Arctic and the years of 1991 to 2019 (Xie et al., 2017; Xie and Bertino, 2022). Detailed quality assessment is available for the EPS version of the Barents-2.5 forecast in 2022 (Röhrs et al., 2023) and 2022 to present (MET-Norway, 2024). However, the validation is not available for all variables, is partly focused on different regions than this study, it is not available for the analysed years of 2020 and 2021 and the non-EPS version of the model. Validation for the sea surface velocity in the Barents-2.5 EPS forecast is further available for the year 2021 and a limited amount of point observations (Idžanović et al., 2023). The Barents-2.5 hindcast is validated for the years 2010-2019 and the model domain (Idžanović et al., 2024).

The comparison to observations in the entire Arctic (Xie and Bertino, 2022), the marginal Barents Sea and point observations (Idžanović et al., 2023) is not representative for the in the complicated spatial characteristics of the Barents Sea. The included years may not be representative for the entire study period, as multi-year variability and changes to due to climate change are large in the Barents Sea. Further, the quality assessment of some of the ocean- and sea ice variables lacks an sufficient amount of independent data (Xie and Bertino, 2022). Sufficiently available observation may have large observational or representative-ness error. Some of the model version are not validated directly, but rely on the validation of related versions. Following, the

 

quality assessments for Topaz and the Barents-2.5 products are difficult to compare and have partly limited applicability for this study.

*Sea surface temperature*

The Topaz $SST$ has a spatial average bias of the Arctic Ocean domain that varies seasonally between $-0.2°$C (winter) and $+0.2°$C (summer) in the years of 2010 to 2014 (Xie et al., 2017; Xie and Bertino, 2022). Reasons for the seasonal varying bias are the seasonal variation in available observations, and in summer, potentially the representation of the mixed layer and the atmospheric radiative forcing (Xie et al., 2017). The spatial validation of Topaz $SST$ (1999-2013) showed a large negative

bias ($< -0.04°C$) between the Barents Sea and the Eurasian Basin, and large positive bias along the shelf-edge of southern Svalbard, Spitsbergen Bank, Storfjorden- and Bjørnøya Trough (Xie et al., 2017). Xie et al. (2017) mentioned problems with the circulation of Atlantic water inflow, the topographic steering and spatially varying observational errors as possible reasons. The Barents-2.5 $SST$ shows a cold bias against observations from Sentinel in 2022 (forecast, EPS) and observations from drifters in 2010-2018 (hindcast) (Idžanović et al., 2024; Röhrs et al., 2023). The hindcast bias is about $1-2°$C in the first half

of the year, smaller in the second half of the year, with little multi-year variation (Idžanović et al., 2024). As a consequence, the Barents-2.5 hindcast shows larger errors than the Topaz reanalysis, especially between January and July. The forecast shows too small $SST$ during most of the year, too large $SST$ (and largest error) in summer in the analysis domain (Röhrs et al., 2023; MET-Norway, 2024). The $SST$ in the Barents-2.5 forecast in 2022 was found to have skill in the Barents Sea, however also large mismatches in the marginal ice zone (Röhrs et al., 2023). Generally too low $SST$ may derive from a coupling to too

extensive sea ice.

*Sea ice concentration*

The $CI$ agrees well in Topaz and the assimilated data from OSI-SAF in 1991-2013 (Xie et al., 2017). The $CI$ in Topaz has a bias of around $-2\%$ in the Arctic Ocean and the years of 2010-2014 (Xie and Bertino, 2022). Although the seasonal and interannual trend is captured well, the spatial average varies between -0.06 (winter) and 0 (summer) seasonally. The sea ice

edge has been found to extent and decline too fast and is related to large errors, especially in March (Xie et al., 2017; Xie and Bertino, 2022). Following also, the sea ice area (with $CI > 15\%$) is too small, compared to the OSI-SAF data. The $CI$ in the Barents-2.5 forecast (EPS, Barents Sea, 2022) has skill for short lead times with data assimilation, however the model state drifts off and looses skill after few days without observations (Röhrs et al., 2023). The comparison with the satellite-based observations of OSI-SAF from 2010-2018 (hindcast), SIRANO (https://cryo.met.no/en/sirano) and the ice chart product

(https://cryo.met.no/en/latest-ice-charts) of MET-Norway for 2022 (forecast) revealed too large $CI$ throughout most of the seasons and years (Röhrs et al., 2023; Idžanović et al., 2024). The results vary significantly with the comparison data and the years. The deviations from the observations is larger for the Barents-2.5 hindcast than the Topaz reanalysis (Idžanović et al., 2024).

*Sea ice thickness*

The sea ice thickness in Topaz is underestimated in the Arctic and the years 2010-2014 with a bias of $-0.4$ to $-0.56\,$m, espe-



cially for thick sea ice (Xie and Bertino, 2022). However, regional differences are large and, for the example of the northern Barents Sea, Topaz underestimates the sea ice thickness in autumn and over- and underestimates in spring (Xie and Bertino, 2022). The thickness bias is even larger (up to $1.1\mathrm{m}$) in the comparison to ICE-SAT (2003-2008) and Operation Icebridge data
(2009-2011) in Xie et al. (2017), however in the used version of the Topaz product the sea ice thickness was not assimilated. The sea ice thickness quality is not evaluated in Barents-2.5.

*Sea ice drift*
The sea ice velocities in Topaz are overestimated in the Arctic from 2010 to 2014, with a mean spatial and temporal bias of
$1.7\,\mathrm{km\,day}^{-1}$, a too small seasonal variation with a spatial average bias of $-12\,\mathrm{km\,day}^{-1}$ (summer) to $+5\,\mathrm{km\,day}^{-1}$ (winter), and largest biases of up to $7-15\,\mathrm{km\,day}^{-1}$ north of Svalbard and Franz-Josef-Land (Xie et al., 2017; Xie and Bertino, 2022). The drift speed and its bias change after the reduction of atmospheric drag coefficient in 2010 (Xie et al., 2017; Xie and Bertino, 2022). The sea ice drift direction has a bias of $20°$ and a RMSE of $30°$ to the right in the Arctic and the years of 1991-2019 (Xie and Bertino, 2022). The sea ice drift in the Barents-2.5 forecast shows some skill in the Barents Sea (EPS, 2022), but is
generally too large in comparison to OSI SAF observations (Röhrs et al., 2023).

*Sea water surface velocity*
In general, the predictability of sea water velocity is limited by the chaotic nature of the system, the availability of observations and information about errors and thus inaccurate initial conditions. Model skills and their representation of individual mech-
anisms vary over time- and spatial scales. Despite partially low model skill, we can still benefit from accurate representation of water velocities that are steered by tides, wind and topography, and ensemble predictions systems supplying robust error statistics. (Röhrs et al., 2023)

The sea water velocity in Topaz has slower water velocity than Barents-2.5 and less spatial gradient, due to its larger
horizontal resolution. No detailed validation or comparison is available. The surface sea water velocity in the Barents-2.5 forecast (EPS, 2021) in the Barents Sea, has low predictive skill for speed and direction (Idžanović et al., 2023). Thereby, ensemble members forced by higher resolution atmospheric data (AROME-Arctic) at the surface yielded better results than members forced by lower resolution atmospheric data (ECMWF Ensemble) (Idžanović et al., 2023). The sea water surface velocity has some skill for events with largely wind-driven water velocities (when the otherwise uncertain initial conditions are
less relevant)(Idžanović et al., 2023). Despite low predictive skill, the water velocity is used for estimating the probability of risk conditions in planning and operations, as it the ensemble provides skilful error statistics and highest available horizontal resolution (Idžanović et al., 2023). The forecast would benefit from more extensive assimilation of local water velocities, salinity and temperature observations (Idžanović et al., 2023).

## A2  Comparative study of ERA5 and CARRA

Two selected atmospheric models are compared based on existing literature, following the structure of Sect. A1.



### A2.1 Model setup

ERA5 consists of the 2-way coupled atmosphere-ocean and atmosphere-land components, that allow for e.g. the atmosphere generating waves, which impact the wind again by surface roughness (Hersbach et al., 2020). For every grid cell surface tiles are defined (e.g. ocean, sea ice or land type) (ECMWF, 2016). CARRA consists of the coupled components of LSMIX,
Canari, SURFEX and 3D-VAR, that mix the large-scale information from ERA5 into the model background, assimilate surface observations and surface information, simulate the surface, and assimilate atmospheric data (Yang et al., 2020a; Køltzow et al., 2022).

### A2.2 Horizontal and temporal resolution

The horizontal resolution is approximately $31\,\mathrm{km}$ in the study domain for ERA5 and widely homogeneous $2.5\,\mathrm{km}$ in the do-
main for CARRA (ECMWF, 2016; Køltzow et al., 2019) (Table 1). The regular latitude-longitude grid in ERA5 (Hersbach et al., 2020) is less suitable in polar regions than the curvilinear grid in Topaz, Barents-2.5 and CARRA (MDS, 2023; Röhrs et al., 2023; Schyberg et al., 2023), as the grid spacing varies largely with latitude, causing inhomogeneous grid spacing and coarse latitudinal resolution. CARRA benefits from a high horizontal resolution of the variables, topography and coastlines. ERA5 is limited in the representation of small scale processes and local gradients due to their coarse resolution.

The temporal resolution is $\mathrm{hourly}$ in ERA5 and $3-\mathrm{hourly}$ in CARRA. ERA5 benefits from an $\mathrm{hourly}$ temporal resolution, that allows the representation of diurnal and fast processes, e.g. short-scale weather processes.

### A2.3 Forcing and boundary conditions

At the surface boundary, ERA5 is forced by $SST$ from ORAS5 and $CI$ from a range of perturbed observation products (e.g.
HadISST2, OSTIA and OSI-SAF at 0.25 to 0.5 deg resolution) Hersbach et al. (2020). At the surface boundary, CARRA prescribes $SST$ and $CI$ from high-resolution satellite products (Yang et al., 2020b). At the lateral boundaries, CARRA is forced by a selection of upper-air and surface variables of an $61\,\mathrm{km}$ resolution-, ensemble- version of ERA5 (Yang et al., 2020b) (Fig. 1).

### A2.4 Data assimilation

ERA5 and CARRA assimilate a wide range of observational and model data, however the assimilation is more extensive for CARRA (Køltzow et al., 2022). Additional assimilated data consist of reprocessed satellite observations (e.g. snow albedo on glaciers, $CI$, scatterometer), local surface observation datasets (e.g. from the Greenland Ice Sheet and measurements from local weather stations, provided by the national weather services). Wind speeds are assimilated in ERA5 from a wide spectrum of observation sources (e.g. radar wind profiler, buoys, ships, radiosondes, aircrafts, and satellite-based infrared-, microwave-
and scatterometer- measurements) (Hersbach et al., 2020). Although a range of atmospheric variables is assimilated from ob-



servations and the $31\,\mathrm{km}$ version of ERA5, 10m wind speed is not assimilated in CARRA and thus independent of observations (Køltzow et al., 2022).

### A2.5 Representation of the Arctic

In comparison to ERA5, CARRA has a higher resolution, improved physical representation of cold surfaces by model tuning and improved regional physiography and orography (Køltzow et al., 2022). For example, CARRA has a corrected land-sea-ice-mask, coastline and elevations data and fractions of glacial coverage on Svalbard (Køltzow et al., 2022; Yang et al., 2020a).

### A2.6 Prognostic variables

10m wind is a prognostic variable in ERA5 and CARRA (Hersbach et al., 2020; Køltzow et al., 2022). Difficulties in the estimation of the 10 m wind lie in complex topography, knowledge of the surface type and the representativity of point observations (from e.g. SYNOP stations) for the model grid cell (ECMWF, 2016). IN ERA5, the surface roughness is based on the respective surface tile types, land cover for land tiles, wind speed over ocean, and ice concentration over sea ice.

### A2.7 Quality of the 10m wind variable

The quality assessment and comparison of 10 m wind in ERA5 and CARRA is restricted to several small regions within the North Atlantic and European Arctic, limiting the applicability to the Barents Sea. ERA and CARRA are both compared to (the same set of) observations, as well with one another. 10m wind speed of ERA5 and CARRA is compared in various studies, e.g. Køltzow et al. (2019), Køltzow et al. (2022) and Giusti (2024). In general, CARRA variables benefit from added value over ERA5 with a smaller spatial and temporal bias, better capturing the spatial and temporal variability and local phenomena, in high impact weather and the climatology (Køltzow et al., 2022; Yang et al., 2020c).

In detail, the 10m wind speed is underestimated (negative bias) in ERA5 and overestimated (positive bias) in CARRA in most of the North Atlantic and European Arctic (Giusti, 2024). Around Svalbard, the bias is $-1.4\,(-1)\,\mathrm{m\,s^{-1}}$ for ERA5 and $+0.5\,(-0.2)\,\mathrm{m\,s^{-1}}$ for CARRA, in winter (summer), averaged over the years of 2010-2019 (Giusti, 2024). Despite a moderate mean error (e.g. for Svalbard) (Giusti, 2024), the spatial variability of systematic and random errors is large (Køltzow et al., 2022).

The temporal variability of the wind speed is underestimated in ERA5 (and represented better in CARRA) (Køltzow et al., 2022). However, ERA5 has slightly decreased standard deviation of errors, due to its larger temporal frequency (hourly compared with $3-\mathrm{hourly}$ in CARRA) (Giusti, 2024). Both reanalyses have an annual cycle with largest errors during winter and smallest error in summer (in comparison to SYNOP observations for Svalbard and other islands in the North Atlantic, from 1998-2020) (Køltzow et al., 2022).



The representation of local phenomena (e.g. marine icing and polar lows), wind speed over complex topography and over sea ice and their local differences is improved in CARRA, but similar on open ocean (where the errors are smaller in both models) (Giusti, 2024; Køltzow et al., 2019, 2022).


10m wind speed is affected by the surface roughness of the varying sea ice representations in ERA5 and CARRA. Both models assume level sea ice (independent of the sea ice thickness) and prescribe sea ice concentration by satellite-based products (Copernicus Climate Change Service (C3S) (2023); Giusti (2024) and Personal communication at CARRA workshop, 21 Sep 2023). Therefore, the near surface wind is overestimated in regions with multiyear ice. However, the improved physical

parametrisation of sea ice and high resolution reprocessed assimilated sea ice satellite data in CARRA add value for variables on sea ice (Giusti, 2024).

CARRA is designed-for and tuned-to the European Arctic. On one side, ERA5 was observed to be performing well in the Arctic (Graham et al., 2019), especially over ocean (Hersbach et al., 2020). On the other side, ERA5 has large problems in representing certain parts of the Arctic domain and their phenomena, such as wind speeds related to polar lows (Køltzow et al.,

830 2022).

Extreme wind speeds are significantly underestimated by ERA5 around Svalbard, and slightly overestimated around other islands of the North Atlantic by both models (Køltzow et al., 2022).

The observational error varies for the two models, as ERA5 (CARRA) does (not) assimilate wind speed observations. The

representativity error (how representative grid cells are for point observations and vice versa) makes up $40 - 55\%$ of the total error in CARRA and $60 - 70\%$ in ERA5, due to the different grid resolution (Køltzow et al., 2022). The sub-grid variability (of 10m wind speed) is partly large (Køltzow et al., 2022). The total error (model, observational- and representativity- and random error) is smaller in CARRA (Køltzow et al., 2022; Giusti, 2024).

*Author contributions.* Data pre-processing, statistical analysis and original draft of manuscript: LH. Supervision during all stages of the

study and review of the manuscript: RKL, KVH.

*Competing interests.* The authors declare that they have no conflict of interest.

*Acknowledgements.* The authors wish to acknowledge the support from the Research Council of Norway through the RareIce project (326834) and the support from all RareIce partners. The authors also acknowledge Edel Rikardsen, Johannes Röhrs and Martina Idžanović (Met-Norway) for providing a preliminary version of the Barents-2.5 hindcast and supporting the process of understanding the product.

The ERA5 (Hersbach et al., 2023) and CARRA data (Schyberg et al., 2023) were downloaded from the Copernicus Climate Change Service (2023). The results contain modified Copernicus Climate Change Service information (2023). Neither the European Commission nor



ECMWF is responsible for any use that may be made of the Copernicus information or data it contains. This study has been conducted using E.U. Copernicus Marine Service Information, (MDS, 2023).



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
