# Peer review of "Comparison of variables from ocean, sea ice and atmosphere models as forcing data for iceberg drift and deterioration models in the Barents Sea in 2010-2014 and 2020-2021 (Part I)."

_EGUsphere, 2024_

## Referee Comment (RC1)

Review of "Comparison of variables from ocean, sea ice, and atmosphere models as forcing data for iceberg drift and deterioration models in the Barents Sea in 2010-2014 and 2020-2021 (Part I)"

**General Comments**

This paper provides a detailed comparison of variables from ocean, sea ice, and atmosphere models (Topaz4b, Barents-2.5, ERA5, and CARRA) in the Barents Sea for the years 2010–2014 and 2020–2021. It establishes a foundational understanding of how these models differ in their representation of key environmental variables. The analysis is aimed at elucidating the potential implications for iceberg drift and deterioration modeling, which are addressed in the sequel paper.

I really appreciate that the Part I is thorough in its statistical analysis and highlights the interplay between model resolution, data assimilation, and physical parameterizations. but I am not sure if the Part I is appropriate and innovative enough because: (1) the paper's reliance on model intercomparison without significant validation against observational data limits its broader applicability; (2) that compromise the future usage of ocean and ice, as well as the atmosphere datasets forcing in the iceberg model because here I don't see some useful suggestions based on the intercomparison. I understand the observations are quite hard to get, but since the focus period is after 2000, which we had lots of in-situ (buoys and moorings both for ocean and sea ice), weather station, airborne, as well as the satellite observations, which can be used in the observations validation. So, the results don't convince me especially when the author find the big differences between Topaz4b and Barents-2.5, but did not mention which is more accurate in representing Barents Sea region in terms of the current state. This lack of clarity becomes a critical issue when reading Part II, where the authors show that iceberg drift and deterioration are highly sensitive to ocean and sea ice forcing. A practical question then arises: which forcing dataset should be used in iceberg modeling? Part I provides no clear answer, which undermines its practical relevance.

Therefore, I recommend that the paper undergo major revisions, at least bring intensive observation validations, before it can be considered for publication.

**Specific Comments:**

1. Line 10: "ERA5 show larger wind speeds", should be "ERA5 shows larger wind speeds".
2. Line 17 and Line 374: "Constraint to the physics and observations" → should be "constrained by the physics and observations"
3. Section 3.1.1: The projection of low-resolution datasets (e.g., 12.3 km in Topaz) onto finer grids using nearest-neighbor interpolation may introduce artifacts, particularly in regions with complex bathymetry. Wouldn't a more refined method, such as inverse distance weighting or bilinear interpolation, yield better results? Additionally, the temporal upsampling of Topaz from daily to 2-hourly intervals might affect the accuracy of comparisons and subsequent model outputs. The authors should critically discuss these potential issues.
4. Temporal resampling: I am uncertain whether the adopted 2-hourly temporal resolution adequately captures short-term dynamics, especially those relevant for iceberg drift. Rapid environmental changes, particularly in regions influenced by tides or strong winds, may not be well-represented at this frequency.
5. Figure 5: The caption for Figure 5 lacks a reference to the color legend, which makes the figure less intuitive for readers.

6. ERA5 and CARRA comparison: The comparison between ERA5 and CARRA seems unnecessary, as CARRA is partly forced by ERA5 surface fields. The dependence between the two datasets inherently limits the differences, as also demonstrated in Part II, where the authors show that atmospheric forcing has a relatively small impact on iceberg drift and deterioration. The practical utility of this comparison is unclear.

In general, while the authors have done an excellent job of identifying systematic differences between the datasets, the paper's reliance on intercomparison without validation is a significant limitation. This issue is compounded by the lack of clear, practical recommendations for selecting forcing datasets for iceberg modeling. Given the substantial sensitivity of iceberg behavior to ocean and sea ice conditions demonstrated in Part II, Part I should ideally offer stronger guidance on which model or dataset is more suitable for specific applications. Including observational validation and addressing potential artifacts from resampling methods would significantly improve the paper's credibility and practical relevance.

---

## Author Comment (AC1)

**Author's response to anonymous review of "Comparison of variables from ocean, sea ice, and atmosphere models as forcing data for iceberg drift and deterioration models in the Barents Sea in 2010-2014 and 2020-2021 (Part I)"**

*General Comments*
*This paper provides a detailed comparison of variables from ocean, sea ice, and atmosphere models (Topaz4b, Barents-2.5, ERA5, and CARRA) in the Barents Sea for the years 2010–2014 and 2020–2021. It establishes a foundational understanding of how these models differ in their representation of key environmental variables. The analysis is aimed at elucidating the potential implications for iceberg drift and deterioration modeling, which are addressed in the sequel paper.*

[1] First of all, we, the authors, would like to thank the reviewer for the thorough review and detailed analysis of the manuscript. The provided summary by the reviewer demonstrates a good insight and provides new perspectives on the paper. With the reviewer's comments, we believe the manuscript will be significantly improved. As example, the conclusions could be revised as follows.

*Revised conclusion:*

*The advantages of using numerical models of atmosphere, ocean and sea ice are numerous, however, the discrepancy in the representation of the Arctic domain in these models cautions engineers and scientists and urges them to carefully evaluate the applicability of these models for the different use-cases. In this study, ocean, sea ice and atmosphere variables in the Barents Sea from the models Topaz, Barents-2.5, ERA5 and CARRA are statistically compared over the years 2010-2014 and 2020-2021. This detailed and case-specific knowledge is complemented by a comparison of the model setups and accuracy from literature. This novel composite of knowledge about the deviations in ocean, sea ice and atmosphere variables is used in the sequel study Herrmannsdörfer et al., 2024) to examine the effects of these deviations on iceberg drift and deterioration simulations in the Barents Sea. The results of this study may also be used in a wider range of geophysical research and applications in the domain (e.g. analysis of sea ice climatology along a shipping route).*

*In the literature, we find similarities and differences in the model setup, forcing data, data assimilation approach, resolution and representation of the bathymetry for the models Topaz, Barents-2.5, ERA5 and CARRA. Further, we find that the models are interconnected by assimilating the respective other models (or different version of a similar setup) as forcing at the ocean, sea ice and atmosphere interface and the lateral boundaries. For example, ERA5 forces CARRA on the lateral boundaries and Topaz provides the initial conditions to the Barents-2.5 hindcast. The comparison of the model setups may give us an understanding of the similarities in the model outputs, inheritance of errors and presents information important for the use in iceberg simulations (e.g. which processes are represented in the sea surface current).*

*Analysing the variables of interest, we confirm and also quantify phenomena that were previously mentioned in the literature but without any quantification, for example, larger CI, hsi and larger coverage of the Barents Sea with (light and heavy) sea ice, in Barents-2.5, compared to Topaz. We set those differences in relation with the known underestimation of CI and hsi in Topaz and general overestimation in Barents-2.5. In addition, we find that temporal characteristics of sea ice growth and decline in the Barents Sea vary in the sea ice models. Compared to the known too fast decline and freeze-up in Topaz, the melt season is delayed, and the sea ice advance is similar in Barents-2.5.*

*We find average lower SST in Barents-2.5, compared to Topaz, that are consistent with the more extensive representation of sea ice in the same model. We also find that spatial and temporal differences of SST in Topaz and Barents-2.5 coincide with the sea ice edge, the bathymetry and ocean currents with warm Atlantic water inflow, agreeing with previously found Topaz bias due to simulating the Atlantic inflow and effects of the bathymetry.*

*Our detailed analysis showed that sea water surface speed and sea ice drift speed are in average larger in Barents-2.5, compared to Topaz, and especially along bathymetric features with reduced water depth in the Barents Sea (for sea water speed), and the sea ice edge (for the sea ice drift). As described in the literature, the sea surface velocity must be treated with great caution, as the general lack of observations, limits the predictive skill of its forecasts, and limits the constriction to observations in reanalyses and forecasts. In contrast to Topaz, Barents-2.5 may still benefit from the representation of tides, the effect of air pressure on the water surface and high spatial and temporal resolution.*

*10m wind in ERA5 and CARRA is well documented in literature and is very similar over ocean surface. However, we find that CARRA has slightly larger wind speeds over water surfaces and ERA5 has larger speeds over sea ice, which may derive from different representation of surface roughness over water and sea ice, or prescription of different CI products. Further, we find large differences in coastal areas, where representation of the topography and land-sea-mask varies. Those differences shall be seen in the light of previously attributed added value of CARRA over ERA5 in complex topography and over sea ice. Differences are in general small, compared to the absolute wind speeds.*

*We find generally similar ocean, sea ice and atmosphere variable differences along the known main iceberg pathways of the Barents Sea. However, for the individual variables, the model differences are partly more/less strongly pronounced. We find that the difference in sea ice representation is distinctly relevant, as most simulated iceberg trajectories and simulation time steps encounter sea ice.*

*We emphasise that this study is limited to the Barents Sea in the years of 2010-2014 and 2020-2021 and existing model validations. Further, we emphasise that we do not rate the suitability of the models as forcing input to iceberg models, as it varies spatially, temporally and for the simulation goal. Instead, this study shall assist the application specific input choice and explain the observed impact in the sequel part. This study may be extended to a larger number of years and variables for the use in other applications.*

Detailed answers to all comments are given below.

*I really appreciate that the Part I is thorough in its statistical analysis and highlights the interplay between model resolution, data assimilation, and physical parameterizations. but I am not sure if the Part I is appropriate and innovative enough because: (1) the paper's reliance on model intercomparison without significant validation against observational data limits its broader applicability; (2) that compromise the future usage of ocean and ice, as well as the atmosphere datasets forcing in the iceberg model because here I don't see some useful suggestions based on the intercomparison. I understand the observations are quite hard to get, but since the focus period is after 2000, which we had lots of in-situ (buoys and moorings both for ocean and sea ice), weather station, airborne, as well as the satellite observations, which can be used in the observations validation.*

[2] We have an understanding for the doubt and scepticism by the reviewer to the novelty and innovation of our study. We attribute this to lack of sufficient clarifications in the manuscript

which should certainly be improved (the revised conclusions above provide a good example of this). In the following, we will try to prove that our study contains sufficient level of novelty and innovation to be published in this journal.

The study has two main goals:
- Part I: The thorough and systematic review/comparison (e.g. setup, variables, accuracy) of different models of ocean, sea ice and atmosphere.
- Part II: The impact that those model differences (e.g. in setup, resolution and variable representation) have on the application of (an exemplary setup of) iceberg drift simulation.

While conducting this study (Part I), which is in fact a preparatory study for Part II, we noticed that the literature lacks systematic and thorough review/comparisons of the existing environmental models, in addition to, descriptions of several aspects of the existing models. For example, there is no comparison of Topaz and Barents-2.5 on hand, to the knowledge of the authors, however it is needed to explain differences in part II of the study. Thus, in part I, we gather information from existing model descriptions and validations and set it into context with our detailed model comparison in the Barents Sea. We also set the information of individual models into context with the other models.  We claim that combining those existing and new detailed analysis and setting those into context is a novel contribution of this study.

We would like to argue that validating the variables of the existing environmental models against observations is beyond the scope of this study. Similar efforts analysing multiple models and variables exist already (e.g. Køltzow, 2019, 2022; Röhrs, 2023; Xie and Bertino 2022). Those efforts are utilised in this study.

We claim that the gathered knowledge herein has not only value for the sequel part (Part II) of this study, but is also intended for a wider range of applications. For example, we quantified the wind differences between ERA5 and CARRA over sea ice, which we not published before, but are important to any kind of lower atmospheric analysis in the ice-covered Arctic. Another example is the previously unavailable comparison of Topaz and Barents-2.5, which may support the model choice in any kind of ocean/sea ice analysis and application in the domain (e.g. analysis of long-term and recent sea ice conditions along a shipping route).

We suggest to highlight in the manuscript the approach to combine existing knowledge and more detailed new analysis, and show how they complement one another to a new, valuable piece of information.

Further, we understand the desire to present useful and straightforward suggestions. With this in mind, we would like to point the attention to the following:
- This study is focused on analysing the effects of varying forcing inputs on the simulated trajectories statistically.  We do not intend to rate the performance of the environmental input or the iceberg model. This argument is also key to the review of PartII. For example, we try to characterise which impact different amounts of sea ice in Topaz and Barents-2.5 have on the spatial distribution of icebergs in the Barents Sea. We do not intend tell which distribution is more accurate (and due to the lack of observations data we cannot).
- There is no doubt that validation against observational iceberg data would provide the key indicator of the "best" input to use.  As we do not intend to rate the best performance, we do not validate the simulations with observed iceberg trajectories. Future studies will concern themselves with rating the performance of the iceberg simulations under varied forcing by comparing to iceberg drift observations.

- Further, it is not easy to make a straightforward and simple suggestion about the suitability of the data for iceberg simulations, due to the fact that the environmental models perform differently well in different regions and times. Although, this is inconvenient for the reader, different ocean, sea ice and atmospheric forcing variables may be differently suitable in different regions, during different time periods, for different variables, iceberg model settings and study goals.
  For example, Topaz and Barents-2.5 show clear advantages and disadvantages and are more/less accurate in different aspects. Topaz is more coarse and is neglecting tides. Barents-2.5 aims at being more accurate with higher temporal and spatial resolution. However, it showed clear overestimation of sea ice. Other variables, such as ocean currents, are highly uncertain in both cases.
  Iceberg models treat the forcing data differently and are differently tuned, which may cause better results with one or the other forcing.
  The suitability of the forcing also depends on the simulation goal (e.g. statistics of iceberg occurrence or individual trajectories) and which processes it should represent (e.g. tides).

The comment indicates a need for a more thorough explanation of the scope and goals of this study in the manuscript. It should highlight why neither a comparison of the environmental data, nor of the simulated iceberg tracks to observations is attempted. It should also explain why a judgment of the suitability in iceberg simulations is not appropriate or intended in this study. We are sorry that an ambiguity in the motivation led to mismatch of the readers expectations

*So, the results don't convince me especially when the author find the big differences between Topaz4b and Barents-2.5, but did not mention which is more accurate in representing Barents Sea region in terms of the current state.*
[3]
- Answer [2] describes why we cannot give an estimate of which environmental models serves best as input to iceberg simulations.
- Answer [2] also explains how this study is innovative. This study describes both the known (but partly unpublished) "big differences" (e.g. more sea ice in Barents-2.5, compared to Topaz), previously unpublished smaller differences (e.g. wind difference over sea ice of ERA5 and CARRA) and spatial and temporal details. Those details are needed to explain the differences in the iceberg simulations in PartII in a detailed way. Some details contradict the average model difference (e.g. more sea ice in Topaz, compared to Barents-2.5, in 2020), highlighting the importance of the on-hand comparison.
- For the case of the ocean current, for example, the lack of observations makes a validation of the variable difficult for all models (as in e.g. Röhrs 2023b, Idzanovic 2024) and we cannot say which model performs best.

*This lack of clarity becomes a critical issue when reading Part II, where the authors show that iceberg drift and deterioration are highly sensitive to ocean and sea ice forcing. A practical question then arises: which forcing dataset should be used in iceberg modeling?*
[4] We understand the desire from straight-forward outcomes. Answer [2] describes why this is not possible.

*Part I provides no clear answer, which undermines its practical relevance.*

[5] The authors believe that the manuscript has practical relevance, despite:
- No new methodology, data or clear suggestions, as presented in answer [2].

The practical relevance of Part I is given by
- the absence of such detailed comparison in the literature (eg. between Topaz and Barents-2.5, see answers [2,3]).
- the need of those information in part II (see answers [2,3]).
- potential relevancy to applications outside the topic of iceberg simulation (see answer [2]).

The main findings are:
- We describe similarities and differences in the model setup, forcing data, data assimilation approach, resolution and representation of the bathymetry of the atmosphere, ocean and sea ice models, as well as their interconnection by various assimilation methods.
- We quantify how Barents-2.5 shows more sea ice in comparison to Topaz, show differences in their temporal characteristics and the connected differences in the sea surface temperature.
- We describe spatial characteristics of larger sea surface velocity in Barents-2.5, compared to Topaz.
- We confirm high similarity of ERA5 and CARRA, but also show spatial differences along the coastlines and over open water and sea ice.

Suggestions about the reformulation of the manuscript are also given in answer [2].

*Therefore, I recommend that the paper undergo major revisions, at least bring intensive observation validations, before it can be considered for publication.*

[6] As described in answer [2], including observations is far outside the scope of this study, but here-indicated changes shall help to present the results and their importance more clearly.

*Specific Comments:*
*1. Line 10: "ERA5 show larger wind speeds", should be "ERA5 shows larger wind speeds".*
*2. Line 17 and Line 374: "Constraint to the physics and observations" → should be "constrained by the physics and observations"*

[7] Thank you. This will be corrected.

*3. Section 3.1.1: The projection of low-resolution datasets (e.g., 12.3 km in Topaz) onto finer grids using nearest-neighbor interpolation may introduce artifacts, particularly in regions with complex bathymetry. Wouldn't a more refined method, such as inverse distance weighting or bilinear interpolation, yield better results? Additionally, the temporal upsampling of Topaz from daily to 2-hourly intervals might affect the accuracy of comparisons and subsequent model outputs. The authors should critically discuss these potential issues.*

[8] We appreciate the critical reflection of the used methodology.
- However, the method recreates the fixed iceberg model setup in Part II, inherited from Monteban (2020). The authors explicitly exclude the analysis and study of potential improvements of the iceberg model settings from this study. Using a more accurate approach in Part I, would not relate to the data assimilation of the iceberg model. This is true for both spatial interpolation and temporal resampling.
- The critical discussion of potential artefacts and limitations due to used methods is a very good idea.
- I therefore suggest that we explain the reasoning behind the method choices more sharply and take focus from the description of those methods, as they may confuse the

reader about the goals and scope of the manuscript. If the reviewer wishes, we can also add a short discussion of potential artifacts and how the choice in methodology may impact or limit the use in other applications.

*4. Temporal resampling: I am uncertain whether the adopted 2-hourly temporal resolution adequately captures short-term dynamics, especially those relevant for iceberg drift. Rapid environmental changes, particularly in regions influenced by tides or strong winds, may not be well-represented at this frequency.*
[9] We agree that hourly input (highest available resolution) may improve the simulation results for individual trajectories.
- As described in answer [8], the iceberg model setup is inherited from Monteban (2020). As the study excludes model optimisation and as the temporal resampling method in Part I should resemble the iceberg model, we resample the data as described.
- We suggest adapting the manuscript as described in answer [8] and mention the possibilities of potential model improvements for future studies.

*5. Figure 5: The caption for Figure 5 lacks a reference to the color legend, which makes the figure less intuitive for readers.*
[10] Thank you for pointing this out. I will add a colour bar label to Figure 5 and other Figures that may miss it.

*6. ERA5 and CARRA comparison: The comparison between ERA5 and CARRA seems unnecessary, as CARRA is partly forced by ERA5 surface fields. The dependence between the two datasets inherently limits the differences, as also demonstrated in Part II, where the authors show that atmospheric forcing has a relatively small impact on iceberg drift and deterioration. The practical utility of this comparison is unclear.*
[11] We understand why the discussion of ERA5 vs CARRA may seem pointless. However, we want to point out following:
- As described, the difference of CARRA and ERA5 is small, but it exists. Although the similarities of those models are well documented, the discovered differences herein over sea ice were not published before, to the knowledge of the authors, highlighting the importance of this comparison.
- Furthermore, small but important differences in the iceberg simulations results due to varied wind forcing are found in part II, e.g. for the spatial iceberg occurrence (Figure 5) and individual trajectories (Figure 7), further highlighting the importance of atmospheric data comparison.
- We therefore suggest keeping the comparison of ERA5 and CARRA, while adding a short explanation of why the comparison is important to the manuscript and the sequel part.

*In general, while the authors have done an excellent job of identifying systematic differences between the datasets, the paper's reliance on intercomparison without validation is a significant limitation. This issue is compounded by the lack of clear, practical recommendations for selecting forcing datasets for iceberg modeling. Given the substantial sensitivity of iceberg behavior to ocean and sea ice conditions demonstrated in Part II, Part I should ideally offer stronger guidance on which model or dataset is more suitable for specific applications. Including observational validation and addressing potential artifacts from resampling methods would significantly improve the paper's credibility and practical relevance.*
[12] Thank you for this compressed summary of comments. We hope the above answers to the comments and suggestions for manuscript changes satisfy the reviewer and editor leading to a successful publication.

References:

- Idžanovi´c, M., Rikardsen, E. S. U., and Röhrs, J.: Forecast uncertainty and ensemble spread in surface currents from a regional ocean model, Frontiers in Marine Science, 10, https://doi.org/10.3389/fmars.2023.1177337, 2023.
- Køltzow, M., Casati, B., Bazile, E., Haiden, T., and Valkonen, T.: An NWP Model Intercomparison of Surface Weather Parameters in the European Arctic during the Year of Polar Prediction Special Observing Period Northern Hemisphere, Weather and Forecasting, 34, 959 – 983, https://doi.org/10.1175/WAF-D-19-0003.1, 2019.
- Køltzow, M., Schyberg, H., Støylen, E., and Yang, X.: Value of the Copernicus Arctic Regional Reanalysis (CARRA) in representing near surface temperature and wind speed in the north-east European Arctic, Polar Research, 41, https://doi.org/10.33265/polar.v41.8002, 2022.
- Monteban, D., Lubbad, R., Samardzija, I., and Løset, S.: Enhanced iceberg drift modelling in the Barents Sea with estimates of the release rates and size characteristics at the major glacial sources using Sentinel-1 and Sentinel-2, Cold Regions Science and Technology, 175, 103 084, https://doi.org/10.1016/j.coldregions.2020.103084, 2020.
- Röhrs, J., Gusdal, Y., Rikardsen, E., Duran Moro, M., Brændshøi, J., Kristensen, N. M., Fritzner, S., Wang, K., Sperrevik, A. K., Idžanovi´c, M., Lavergne, T., Debernard, J., and Christensen, K. H.: Barents-2.5km v2.0: An operational data-assimilative coupled ocean and 910 sea ice ensemble prediction model for the Barents Sea and Svalbard, Geoscientific Model Development Discussions, 2023, 1–31, https://doi.org/10.5194/gmd-2023-20, 2023.
- Röhrs, J., Sutherland, G., Jeans, G., Bedington, M., Sperrevik, A., Dagestad, K.-F., Gusdal, Y., Mauritzen, C., Dale, A., and LaCasce, J.: Surface currents in operational oceanography: Key applications, mechanisms, and methods, Journal of Operational Oceanography, 16, 60–88, https://doi.org/10.1080/1755876X.2021.1903221, 2023.
- Xie, J. and Bertino, L.: Quality infromation Document - Arctic Physical Multi Year Product ARCTIC_MULTIYEAR_PHY_002_003, Tech. rep., E.U. Copernicus Marine Service Information (CMEMS). Marine Data Store (MDS), https://doi.org/10.48670/moi-00007, 2022.